# Structural insights into transcriptional regulation by the helicase RECQL5

Alfredo Jose Florez Ariza [1,2,7], Nicholas Z. Lue [1,7], Patricia Grob [1,3], Benjamin Kaeser[4], Jie Fang[3], Susanne A. Kassube [2,6] & Eva Nogales [1,3,4,5] ✉

Transcription poses a major challenge for genome stability. The RECQL5 helicase helps safeguard genome integrity and is the only member of the human RecQ helicase family that directly binds to RNA polymerase II (Pol II) and affects its progression. While RECQL5 mitigates transcription stress in cells, the molecular mechanism underlying this phenomenon is unclear. Here, we use cryo-electron microscopy to determine the structures of stalled human Pol II elongation complexes (ECs) bound to RECQL5. Our structures reveal the molecular interactions stabilizing RECQL5 binding to the Pol II EC and highlight its role as a transcriptional roadblock. Additionally, we find that, in its nucleotide-free state, RECQL5 twists the downstream DNA in the EC and, upon nucleotide binding, undergoes a conformational change that allosterically induces Pol II toward a post-translocation state. We propose that this mechanism may help restart Pol II elongation and, therefore, contribute to reducing transcription stress.

Despite its fundamental importance for life, transcription has genome-destabilizing effects that pose challenges for cells[1]. Collisions between transcribing RNA polymerases (Pols) and the replication machinery, for example, can stall replication forks and are well-known sources of DNA damage[1–3]. Cells, therefore, require mechanisms to resolve such conflicts and mitigate the deleterious side effects of transcription. RecQ helicases are key players in these efforts to safeguard genome stability. In humans, the RecQ family comprises five 3′-to-5′ DNA helicases: RECQL1, BLM (Bloom syndrome helicase), WRN (Werner syndrome helicase), RECQL4 and RECQL5 (refs. 3,4). Testifying to the importance of this family, mutations in several RecQ genes—*BLM*, *WRN* and *RECQL4*—are associated with human genetic disorders marked at a cellular level by erosion of genome integrity[3–7]. Although RECQL5 has not been conclusively linked to human diseases, loss of RECQL5 is known to promote cancer in mice, likely as a result of increased genome instability[8]. Consistently, there is some evidence linking RECQL5 loss of function to breast cancer susceptibility in humans[9]. On the other hand, RECQL5 has also been shown to be important for triple-negative breast cancer cell growth, as it counters excessive DNA damage arising from replication stress in these cells[10]. These findings underscore the importance of RECQL5's multifaceted roles within the cell.

RECQL5 has been proposed to protect cells against genome instability through several mechanisms. It has long been known that RECQL5 can remove filaments of RAD51 from single-stranded DNA through its helicase activity, thereby suppressing inappropriate homologous recombination[8,11,12]. This disassembly of RAD51 filaments is also a crucial step in the resolution of stalled replication forks and is important for proper replication restart after transcription–replication conflicts (TRCs)[13–15]. Among the RecQ helicases, RECQL5 is also unique in that it directly interacts with Pol II (refs. 16–20). Through this interaction, RECQL5 inhibits transcription both in vitro and in cells[17,21–23]. In cells, loss of RECQL5 has been shown to promote faster transcription but with both greater transcription stress (Pol II pausing, stalling and arrest) and increased rates of chromosomal rearrangements[23]. These results indicate that RECQL5 enhances transcriptional robustness but the mechanistic basis of this function has not yet been elucidated.

[1]California Institute for Quantitative Biosciences (QB3), University of California, Berkeley, Berkeley, CA, USA. [2]Biophysics Graduate Group, University of California, Berkeley, Berkeley, CA, USA. [3]Howard Hughes Medical Institute, University of California, Berkeley, Berkeley, CA, USA. [4]Department of Molecular and Cell Biology, University of California, Berkeley, Berkeley, CA, USA. [5]Molecular Biophysics and Integrated Bioimaging Division, Lawrence Berkeley National Laboratory, Berkeley, CA, USA. [6]Present address: Department of Biochemistry, Universität Zürich, Zurich, Switzerland. [7]These authors contributed equally: Alfredo Jose Florez Ariza, Nicholas Z. Lue. ✉e-mail: enogales@lbl.gov

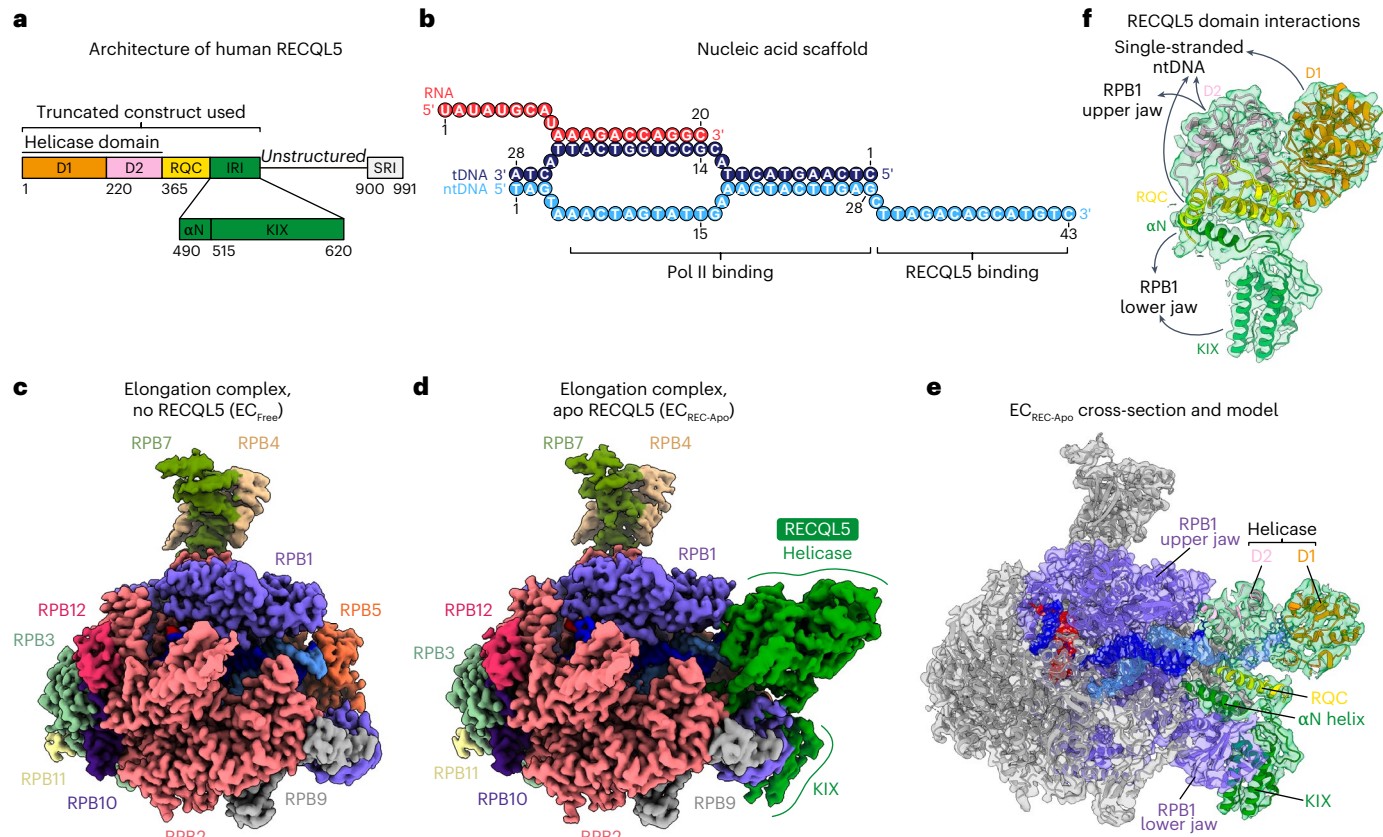

**Fig. 1 | Structure of the Pol II EC bound to nucleotide-free RECQL5. a**, Domain architecture of RECQL5 indicating the truncated construct used in this study (residues 1–620). The helicase domain (comprising the D1 and D2 subdomains), RQC domain, IRI module (comprising the αN helix and KIX domain) and SRI domain are indicated. **b**, Schematic depicting the DNA–RNA scaffold used in this study. Numbers indicate the nucleotide position along each strand. **c**, Cryo-EM map of free Pol II EC (EC_Free). Pol II subunits are colored as indicated by the labels and nucleic acids are colored as in **b**. **d**, Cryo-EM map of Pol II EC bound to

RECQL5_{1–620}-D157A (EC_REC-Apo). Colors are the same as in **c**, with RECQL5 colored in green. **e**, Cross-section of the EC_REC-Apo cryo-EM map (shown in transparency) with the fitted atomic model (shown in ribbon representation). Pol II is shown in gray apart from the RPB1 subunit shown in purple. The RECQL5 model is colored by domain as in **a** and the nucleic acids are colored as in **b**. **f**, View of the cryo-EM density for RECQL5 from EC_REC-Apo (shown in transparency) with the fitted atomic model (shown in ribbon representation), indicating the contacts each domain makes with the EC. Colors are the same as in **e**.

In our previous work, we showed that RECQL5's internal Pol II-interacting (IRI) module, which consists of the αN helix and KIX domain (Fig. 1a), is critical for its interaction with Pol II (ref. 22). This work revealed that RECQL5's KIX domain competes with the transcription elongation factor TFIIS for interaction with Pol II's RPB1 subunit, thereby inhibiting Pol II progression through pause sites. Our low-resolution structure (~13 Å) also showed that RECQL5's helicase domain binds to downstream DNA but was insufficient to define the molecular interactions underlying RECQL5's ability to serve as a roadblock for Pol II advancement. In the decade since this initial study, no other structures have been reported on RECQL5 in the context of transcriptional regulation, emphasizing the need for further detailed structural study.

Understanding the molecular mechanism of RECQL5 transcriptional regulation requires a comprehension of the structural basis of Pol II transcription. During transcription, incorporation of a nucleotide into the growing RNA chain by Pol II generates an active site configuration referred to as a pretranslocation state[24]. Pol II then translocates forward, generating a post-translocation state in which a new free binding site is available for the next incoming nucleotide. Multiple studies, using both X-ray crystallography and cryo-electron microscopy (cryo-EM), have elucidated the structures of Pol II in different translocation states[25–28]. Moreover, a crystallographic structure of Pol II bound to the transcriptional inhibitor α-amanitin showed that Pol II can also adopt a translocation intermediate conformation in addition to these two discrete structural states[29].

Here, we use cryo-EM to determine the structures of Pol II elongation complexes (ECs) engaged with RECQL5 to gain molecular insight into RECQL5 transcriptional regulation. The improvement in resolution over our previous RECQL5–Pol II structure enables us to visualize the specific molecular contacts RECQL5 establishes with the Pol II EC. We find that RECQL5 has the ability to perturb Pol II's translocation state, suggesting a mechanical role in helping restart stalled Pol II.

## Results

### Structure of the Pol II EC bound to nucleotide-free RECQL5

To study the molecular basis for RECQL5's inhibitory effect on transcription, we conducted single-particle cryo-EM analysis of an in vitro reconstituted stalled human Pol II EC bound to RECQL5. The EC was assembled using a nucleic acid scaffold comprising both template and nontemplate strand DNAs (tDNA and ntDNA, respectively) and a hybridized RNA, mimicking a transcription bubble[22,25] (Fig. 1b). Additionally, the scaffold contained an extended single-stranded DNA region (3′ end of the ntDNA) around the Pol II DNA entry site. This scaffold provided a platform for RECQL5 to bind in a head-to-head orientation with respect to Pol II (ref. 22). For our structural studies, we used a truncated RECQL5 construct (RECQL5_{1–620}) encompassing the helicase domain (D1 and D2 subdomains), RecQ C-terminal (RQC) domain and the IRI module (comprising the αN helix and KIX domain) (Fig. 1a). RECQL5's unstructured region and C-terminal Set2–Rpb1-interacting (SRI) domain were not included in this construct. Purified and mildly crosslinked EC bound

## Table 1 | Cryo-EM data collection, refinement and validation statistics

| | EC$_{Free}$ | EC$_{REC-Apo}$ | EC$_{REC-Apo (IRI-focused)}$ | EC$_{REC-AMPPNP}$ | EC$_{REC-ADP}$ |
|---|---|---|---|---|---|
| | (EMD-48071), (PDB 9EHZ) | (EMD-48073), (PDB 9EI1) | (EMD-48074), (PDB 9EI2) | (EMD-48075), (PDB 9EI3) | (EMD-48076), (PDB 9EI4) |
| **Data collection and processing** | | | | | |
| Magnification | ×81,000 | ×81,000 | ×81,000 | ×81,000 | ×81,000 |
| Voltage (kV) | 300 | 300 | 300 | 300 | 300 |
| Electron exposure (e⁻ per Å²) | 50 | 50 | 50 | 50 | 50 |
| Defocus range (μm) | −0.8 to −1.8 | −0.8 to −1.8 | −0.8 to −1.8 | −0.8 to −1.8 | −0.8 to −1.8 |
| Pixel size (Å) | 1.05 | 1.05 | 1.05 | 1.048 | 1.048 |
| Symmetry imposed | $C_1$ | $C_1$ | $C_1$ | $C_1$ | $C_1$ |
| Initial particle images (no.) | 871,524 | 871,524 | 871,524 | 804,990 | 1,048,486 |
| Final particle images (no.) | 174,428 | 24,323 | 103,214 | 80,622 | 17,442 |
| Map resolution (Å) | 2.6 | 3.2 | 2.8 | 3.1 | 3.7 |
| FSC threshold | 0.143 | 0.143 | 0.143 | 0.143 | 0.143 |
| Map resolution range (Å) | 2.4–6.4 | 2.9–7.0 | 2.4–5.6 | 2.9–8.3 | 3.2–8.4 |
| **Refinement** | | | | | |
| Initial model used (PDB code) | 5FLM | 5FLM | 5FLM | 5FLM | 5FLM |
| Model resolution (Å) | 2.7 | 3.3 | 3.4 | 3.4 | 3.9 |
| FSC threshold | 0.5 | 0.5 | 0.5 | 0.5 | 0.5 |
| Model resolution range (Å) | Not applicable | Not applicable | Not applicable | Not applicable | Not applicable |
| Map sharpening $B$ factor (Å²)ᵃ | −51.4 | −10 | −20 | −20 | −20 |
| Model composition | | | | | |
| Nonhydrogen atoms | 32,125 | 36,623 | 2,182 | 36,621 | 35,854 |
| Protein residues | 3,906 | 4,456 | 267 | 4,448 | 4,369 |
| Ligands | 9 | 9 | 0 | 10 | 10 |
| $B$ factors (Å²) | | | | | |
| Protein | 54.86 | 67.56 | 65.48 | 70.85 | 53.66 |
| Ligand | 77.39 | 105.4 | Not applicable | 122.66 | 77.3 |
| R.m.s.d. | | | | | |
| Bond lengths (Å) | 0.004 | 0.004 | 0.004 | 0.004 | 0.008 |
| Bond angles (°) | 0.75 | 0.793 | 0.576 | 0.778 | 0.672 |
| **Validation** | | | | | |
| MolProbity score | 2.44 | 2.28 | 2.60 | 2.24 | 2.64 |
| Clashscore | 11.7 | 15.42 | 11.31 | 16.77 | 13.62 |
| Poor rotamers (%) | 4.45 | 3.51 | 4.96 | 1.28 | 5.13 |
| Ramachandran plot | | | | | |
| Favored (%) | 94.75 | 93.36 | 91.64 | 93.36 | 92.86 |
| Allowed (%) | 4.99 | 6.08 | 7.60 | 6.07 | 6.87 |
| Disallowed (%) | 0.26 | 0.56 | 0.76 | 0.57 | 0.27 |

ᵃThe final sharpening for all the maps was performed using DeepEMhancer as detailed in Methods.

to RECQL5 (catalytically inactive D157A mutant and without addition of adenosine triphosphate (ATP), EC$_{REC-Apo}$) was deposited on graphene oxide grids and subjected to cryo-EM imaging.

Our cryo-EM processing showed the coexistence of both free and RECQL5-bound ECs in our sample. By selecting particles without RECQL5, we were able to solve the structure of a free EC containing only Pol II and the nucleic acid scaffold at 2.6-Å overall resolution (EC$_{Free}$; Fig. 1c, Table 1 and Extended Data Figs. 1 and 2a,b). For the RECQL5-containing particles, we observed extensive flexibility in RECQL5, especially in its helicase domain (Extended Data Figs. 3 and 4a,b). This finding is in accordance with our previous observations, based on low-resolution negative-stain EM[22], that RECQL5's helicase

domain can occupy a range of positions spanning up to a 60° arc around the downstream double-stranded DNA (dsDNA). To improve the resolution in the RECQL5 helicase domain and more clearly visualize its contacts with Pol II, we used a data processing approach in which three-dimensional (3D) classification in a mask focused on the helicase domain was performed on signal-subtracted particles (Methods and Extended Data Fig. 3). For each conformation of the helicase domain identified in this manner, particle subtraction was reverted and the structure of the full complex was refined. This approach proved successful in addressing the conformational heterogeneity present in our dataset and yielded the structure of EC$_{REC-Apo}$ at 3.2-Å overall resolution (Fig. 1d, Table 1 and Extended Data Fig. 2c). The quality of our cryo-EM

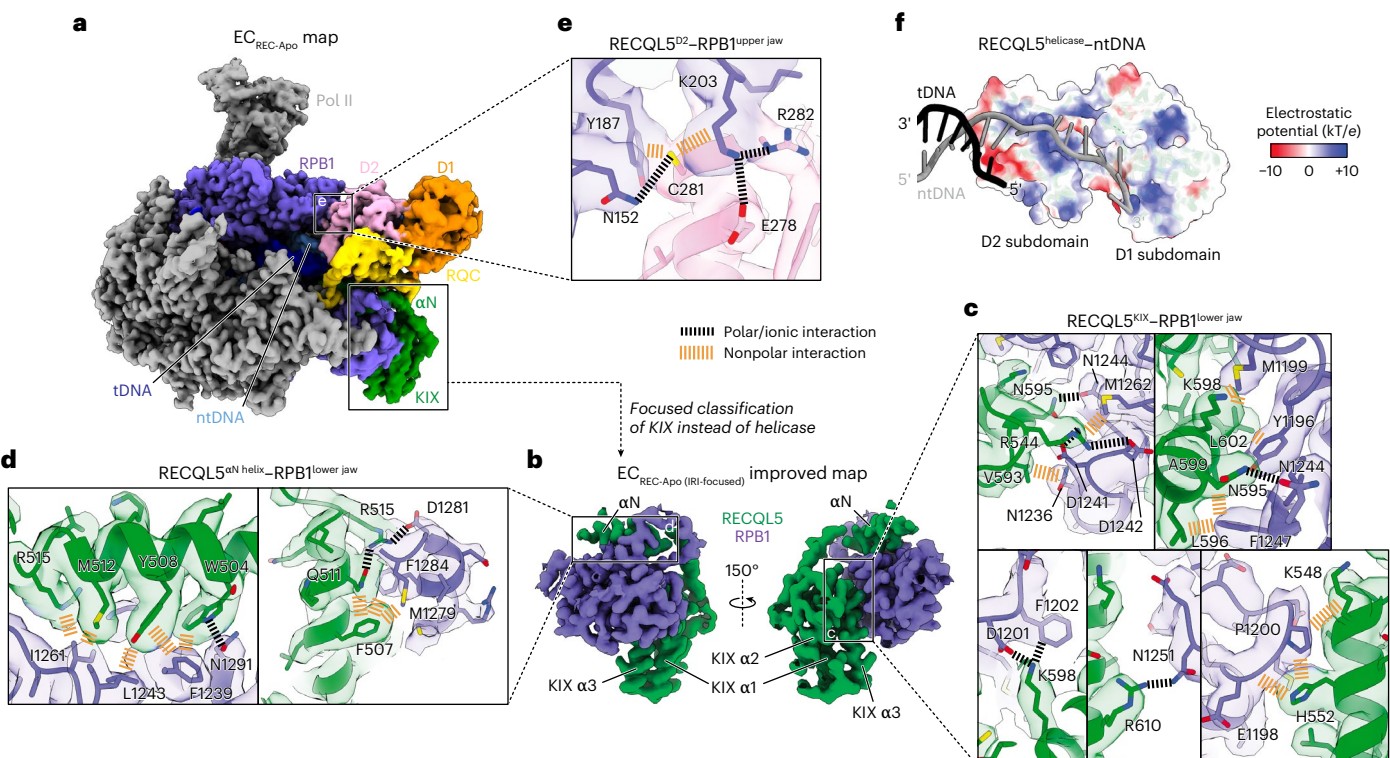

**Fig. 2 | RECQL5 contacts the Pol II EC through multiple domains. a**, Full cryo-EM map for EC_REC-Apo highlighting the Pol II RPB1 subunit (purple) and the different domains in RECQL5 (colored as in Fig. 1a). **b**, Improved cryo-EM map (EC_REC-Apo (IRI-focused)) of the RPB1 lower jaw region engaged with the RECQL5 IRI module (αN helix and KIX domain) (details in Extended Data Fig. 5). **c–e**, Details of the interfaces between RECQL5 and Pol II in EC_REC-Apo, with the cryo-EM map shown as a transparent surface and colored according to the fitted atomic model shown in ribbon and with key residues shown in stick representation. Salt bridges and polar interactions are indicated in black and nonpolar interactions are indicated in orange. Zoomed-in views of the map in **b** at the interfaces between the RPB1 lower jaw and the RECQL5 KIX domain (**c**) or the αN helix (**d**). Zoomed-in view of the map in **a** at the interface between the RPB1 upper jaw and the RECQL5 D2 helicase subdomain (**e**). **f**, View of the interface between the RECQL5 helicase domain and the single-stranded DNA. The tDNA (black) and ntDNA (gray) are shown in cartoon representation, while the helicase domain is shown as a semitransparent surface colored by electrostatic potential.

maps enabled us to model all domains of RECQL5 present (helicase domain, RQC domain and IRI module), in addition to Pol II and the nucleic acid scaffold (Fig. 1e,f).

### RECQL5 contacts the Pol II EC through multiple domains

Our EC_REC-Apo structure shows how RECQL5 engages with the EC (Fig. 1d,e). There are multiple points of contact between RECQL5 and Pol II mediated by both the IRI module (αN helix and KIX domain) and the helicase D2 subdomain (Figs. 1f and 2a). We focused first on the IRI module interactions. To improve the resolution in this region, we reprocessed this dataset, classifying in the IRI module region instead of the helicase domain (Extended Data Fig. 5). This approach greatly improved the map's local resolution at the IRI–Pol II interface, enabling us to trace the polypeptide chain (Fig. 2b–d, Table 1 and Extended Data Fig. 2d). The KIX domain binds to the RPB1 lower jaw at the site predicted previously on the basis of its homology to TFIIS[22,30] and is anchored through interactions mediated by its α1 and α3 helices (Fig. 2c). Several residues in α3 (N595, K598 and R610) participate in hydrogen-bonding or ionic interactions with residues in RPB1. Additionally, residues in the top half of α3 (V593, L596 and L602) and in α1 (K548 and H552) mediate nonpolar interactions with RPB1. R544, which is situated in a loop N-terminal to α1, forms salt bridges with D1241 and D1242 in RPB1. Our structure further shows that the αN helix binds to a shallow nonpolar groove in the top face of the RPB1 lower jaw (Fig. 2d). Several residues (W504, Y508, M512 and R515) stick into this groove to interact with RPB1, while two others (F507 and Q511) interact with residues in the wall of the groove.

The isolated helicase domain of RECQL5 was shown not to bind Pol II in an in vitro pulldown assay[22]. However, our present structure reveals

a direct interaction between the RPB1 upper jaw and the D2 helicase subdomain in the context of the EC (Fig. 2a). This interface is stabilized by interactions between RECQL5 E278 and R282 and RPB1 K203, as well as by both polar and nonpolar interactions between RECQL5 C281 and RPB1 N152, Y187 and K203 (Fig. 2e). We note that the helicase conformation we describe in the EC_REC-Apo structure (stabilized in part by D2–Pol II interactions), which we obtained after rounds of local classification, likely represents the most stable of the many possible poses that it can adopt. Lastly, the EC_REC-Apo structure shows that both helicase D1 and D2 subdomains of RECQL5 bind to the single-stranded ntDNA extension (Fig. 2f) through a positively charged channel.

Comparing our EC_REC-Apo structure to previously determined Pol II structures[31,32] revealed RECQL5's compatibility with other Pol II-associated factors. With respect to elongation factors, it was previously shown that RECQL5 blocks TFIIS[22]. By contrast, RECQL5 can bind to Pol II simultaneously with elongation factors Spt4 and Spt5, showing apparent minor steric clashes with Elf1 (Extended Data Fig. 6a). With respect to initiation factors, RECQL5 exhibits a major steric clash with the XPB subunit of TFIIH in the context of the preinitiation complex (Extended Data Fig. 6b), indicating that RECQL5 binding to Pol II is not compatible with full preinitiation complex assembly. Taken together, our structural analysis reveals the details of how RECQL5 contacts the Pol II EC and its compatibility with other factors important for transcription.

### Free and RECQL5-bound Pol II translocation intermediates

Given that RECQL5 contacts both Pol II and the extruding ntDNA, we considered the possibility that it might affect the Pol II translocation

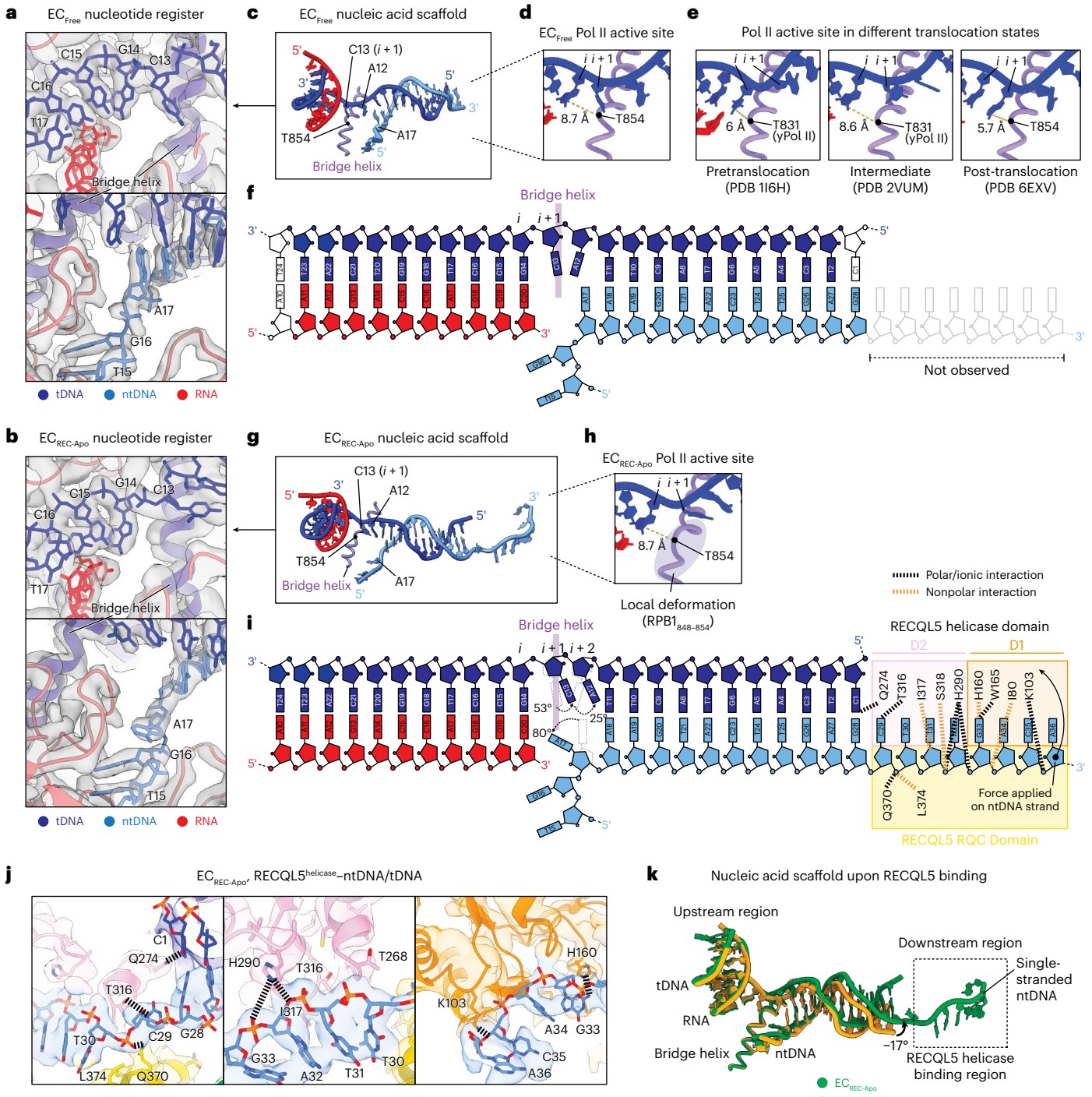

**Fig. 3 | Free and RECQL5-bound Pol II exist in translocation intermediate states. a,b,** Regions near the Pol II active site upstream (top) and downstream (bottom) of the bridge helix in the EC_Free (**a**) and EC_REC-Apo structures (**b**). Cryo-EM density is shown as a transparent gray surface, with the fitted atomic model in ribbon representation and the DNA bases in stick representation. **c,** Nucleic acid scaffold and Pol II bridge helix in EC_Free. **d,** Close-up view of the EC_Free Pol II active site highlighting the distance between the *i* site nucleotide base and the RPB1 T854 residue in the bridge helix. **e,** Views of the Pol II active site in ECs where Pol II is in a pretranslocation conformation (PDB 1I6H, left), translocation intermediate conformation (PDB 2VUM, middle) or post-translocation conformation (PDB 6EXV, right). The distance from the active site nucleotide (*i* or *i* + 1) and the RPB1 T854 residue (T831 in yeast RPB1) is indicated in each case. **f,** Schematic depicting the nucleic acid scaffold in EC_Free highlighting resolved nucleotides and the position of the bridge helix. **g,** Nucleic acid scaffold and Pol II bridge helix in

EC_REC-Apo. **h,** Close-up view of the EC_REC-Apo Pol II active site highlighting the distance between the *i* site nucleotide base and the RPB1 T854 residue in the bridge helix. **i,** Schematic depicting the nucleic acid scaffold in EC_REC-Apo, highlighting resolved nucleotides, the position of the bridge helix and residues in the RECQL5 helicase domain that interact with the nucleic acids. Salt bridges and polar interactions are indicated in black and nonpolar interactions are indicated in orange. **j,** Views of the downstream region in EC_REC-Apo highlighting molecular interactions between the RECQL5 helicase domain and the ntDNA. The cryo-EM map is shown as a transparent surface, with the fitted atomic model in ribbon representation and the residue side chains and DNA bases in stick representation. Salt bridges and polar interactions are indicated with black dashed lines. Residues of interest are explicitly labeled. **k,** Superposition of the nucleic acid scaffolds from the EC_Free (yellow) and EC_REC-Apo (green) structures.

state in the EC. Therefore, we examined the Pol II active site in the EC$_{Free}$ versus EC$_{REC-Apo}$ complexes. The good local resolution (2.3–3.1 Å) attained for the Pol II core region allowed us to assign unambiguously the DNA–RNA register (Fig. 3a,b). In EC$_{Free}$, the tDNA $i$ site base G14 is positioned a distance of 8.7 Å from the RPB1 bridge helix (taking Cα of RPB1 T854 as a reference point) (Fig. 3c,d). This distance is extremely close to the 8.6-Å distance observed for Pol II bound to the transcription inhibitor α-amanitin, which adopts a translocation intermediate conformation[29] (Protein Data Bank (PDB) 2VUM) and is substantially different from the 6-Å distance observed for Pol II in a pretranslocation state[33] (PDB 1I6H) (Fig. 3e). Moreover, when superimposed on the bridge helix, EC$_{Free}$ showed great similarity to the Pol II–α-amanitin structure (root-mean-square deviation (r.m.s.d.) = 0.5 Å) (Extended Data Fig. 6c). These observations indicate that, in the absence of RECQL5, Pol II adopts an intermediate conformation between the pretranslocation and post-translocation states (Fig. 3f). This feature may be because of the specific nucleic acid scaffold we used. For consistency and to enable comparisons, we used the same nucleic acid scaffold design to obtain all cryo-EM structures in this study.

Compared to EC$_{Free}$, EC$_{REC-Apo}$ displays a slightly reorganized Pol II active site (Fig. 3g). In EC$_{REC-Apo}$, the $i$ site tDNA base G14 appears 8.7 Å away from the bridge helix (Fig. 3h), similar to its positioning in EC$_{Free}$. However, we observe a local distortion of the bridge helix in EC$_{REC-Apo}$, in contrast to the EC$_{Free}$ and Pol II–α-amanitin structures (Fig. 3h and Extended Data Fig. 6c). Moreover, while we did not observe density in EC$_{Free}$ corresponding to the single-stranded ntDNA (indicating that it is not stably tethered to Pol II), in EC$_{REC-Apo}$, this region is stabilized through contacts with the RECQL5 helicase and RQC domains (Figs. 2f and 3i,j). Within the helicase domain, most interactions with the single-stranded DNA are mediated by residues in the D2 subdomain, with fewer interactions contributed by the D1 subdomain. In this interaction mode, the RECQL5 helicase appears to adopt a 'pushing and unwinding' state where it pushes the single-stranded DNA and concomitantly bends the downstream dsDNA end by −17°, leading to its partial unwinding (Fig. 3k). This local DNA bending causes distortions of the B-DNA helix structure that are propagated upstream until reaching the DNA–RNA hybrid region, which itself does not display structural rearrangements. Interestingly, when inspecting the Pol II active site in the EC$_{REC-Apo}$ structure, the tDNA bases C13 ($i$ + 1 site) and A12 ($i$ + 2 site) appear greatly rotated toward the downstream region, by 53° and 25° relative to their positioning in the EC$_{Free}$, respectively (Fig. 3i). Additionally, the ntDNA base A17, located immediately downstream of the bridge helix, appears rotated 80° toward the upstream region relative to its position in EC$_{Free}$. Altogether, these observations indicate that Pol II in the EC$_{REC-Apo}$ structure also adopts an intermediate conformation similar to that observed in EC$_{Free}$, albeit with a slightly altered active site configuration.

### Structures of ECs with nucleotide-bound RECQL5 helicase

In light of these observations, we considered whether the Pol II translocation state might change depending on the nucleotide-binding state of the RECQL5 helicase domain. Previous studies showed that RECQL5's helicase activity is not necessary to inhibit Pol II transcription in vitro[21,22]. However, structural work showed that the helicase domain changes conformation upon adenosine diphosphate (ADP) binding, with the D1 subdomain rotating ~20° relative to the D2 subdomain[34]. Those structures were obtained in the absence of DNA; therefore, it remains unclear whether this conformational change can take place in the context of an EC. In turn, we wondered whether a helicase conformational change could affect the translocation state of Pol II.

To investigate these questions, we assembled and purified RECQL5 ECs with a pulldown approach, using wild-type RECQL5 in the presence of either adenylyl-imidodiphosphate (a nonhydrolyzable ATP analog also known as AMPPNP) or ADP (Extended Data Fig. 7a,b). We collected cryo-EM data for these complexes and used a similar processing workflow as for EC$_{REC-Apo}$ to solve their structures, which we refer

to as EC$_{REC-AMPPNP}$ and EC$_{REC-ADP}$ (Table 1 and Extended Data Figs. 2a,e,f, 7c–e, 8 and 9). Comparison of these structures to the EC$_{REC-Apo}$ structure showed that the IRI modules were almost identical in conformation (Extended Data Fig. 7f), with r.m.s.d. less than 1 Å (0.554 Å for EC$_{REC-Apo}$ versus EC$_{REC-AMPPNP}$ and 0.852 Å for EC$_{REC-Apo}$ versus EC$_{REC-ADP}$). These observations support the notion that the IRI module serves to anchor RECQL5 to Pol II throughout the helicase domain's nucleotide binding and hydrolysis cycle.

### RECQL5 nucleotide binding leads to post-translocation Pol II

AMPPNP binding induced noticeable conformational changes within both the Pol II active site and the helicase–ntDNA interaction region. As with the previous structures, we were able to confidently assign the DNA–RNA register in EC$_{REC-AMPPNP}$ and, therefore, identify the locations of the $i$ and $i$ + 1 nucleotides in the Pol II active site (Fig. 4a–c). We observed several major conformational changes in EC$_{REC-AMPPNP}$ compared to EC$_{REC-Apo}$. First, in the tDNA strand, the A12 base ($i$ + 2 site) appears rotated by 16° in the upstream direction, while the ntDNA base A17 is rotated by 92° in the downstream direction (Fig. 4d). Strikingly, the tDNA base C13 ($i$ + 1 site) displayed a large 93° rotation toward the upstream region to be located immediately upstream of the bridge helix, positioned 5.4 Å away from it (Fig. 4c). Comparing this EC$_{REC-AMPPNP}$ structure (Fig. 4c) to previous ECs (Fig. 3e and Extended Data Fig. 6c), we conclude that Pol II adopts a post-translocation conformation in this complex.

To understand how the post-translocation conformation is promoted by AMPPNP binding, we examined the RECQL5 helicase domain's interactions with the single-stranded ntDNA. In EC$_{REC-AMPPNP}$, the single-stranded ntDNA is contacted by additional residues compared to EC$_{REC-Apo}$ (Figs. 3i and 4d,e). Moreover, the helicase D1 subdomain is rotated +6° around the D2 subdomain relative to EC$_{REC-Apo}$ (Fig. 4f), resulting in a new interaction mode that we refer to as a 'pulling and rewinding' state. Here, the RECQL5 helicase appears to pull the single-stranded ntDNA around its 3′ end, concomitantly bending the dsDNA downstream region by +7° and leading to its partial rewinding (Fig. 4g). We also analyzed the evolutionary conservation of the helicase domain to assess the importance of the ntDNA-interacting residues we identified. Many of the residues mediating salt bridges or polar interactions with the ntDNA in our EC$_{REC-Apo}$ or EC$_{REC-AMPPNP}$ structures are highly conserved among RECQL5 orthologs from diverse organisms (S102, K103, E131, W165, H167, R267, H290 and T316), supporting their functional importance (Extended Data Fig. 10). Two of these residues, K103 and R267, are conserved across all animal RECQL5 orthologs but much less conserved among other RecQ family helicases in humans or more distantly related organisms such as *Saccharomyces cerevisiae* (SGS1) or *Escherichia coli* (recQ), suggesting a key role unique to RECQL5.

We also examined the effects of RECQL5 binding to ADP. The cryo-EM structure of EC$_{REC-ADP}$ showed an overall architecture similar to EC$_{REC-Apo}$ and EC$_{REC-AMPPNP}$ (Fig. 1d,e and Extended Data Fig. 7c,d). In EC$_{REC-ADP}$, neither the αN helix nor the KIX domain show major changes relative to their positions in EC$_{RECApo}$ or EC$_{REC-AMPPNP}$ (Extended Data Fig. 7f). The main difference with respect to the two other states analyzed is that the RECQL5 helicase domain in EC$_{REC-ADP}$ exhibits high orientational flexibility relative to Pol II, which led to lower local resolution of this region (7–8.4 Å; Extended Data Fig. 2f), despite extensive data analysis (Extended Data Fig. 9). Moreover, the density assigned to the D1 subdomain remained blurrier than that for the D2 subdomain (Extended Data Fig. 7d), allowing only partial modeling of D1 and the single-stranded ntDNA it interacts with. These results suggest that, in the ADP state, the RECQL5 helicase D1 and D2 subdomains may be more dynamic. Notably, we did not observe as large of a rotation between these domains as seen in crystallographic structures of the free RECQL5 helicase domain[34]. We posit that this may be because of constraints imposed by DNA and Pol II binding within the context of the EC. Inspection of the Pol II active site in EC$_{REC-ADP}$ revealed that Pol II is in

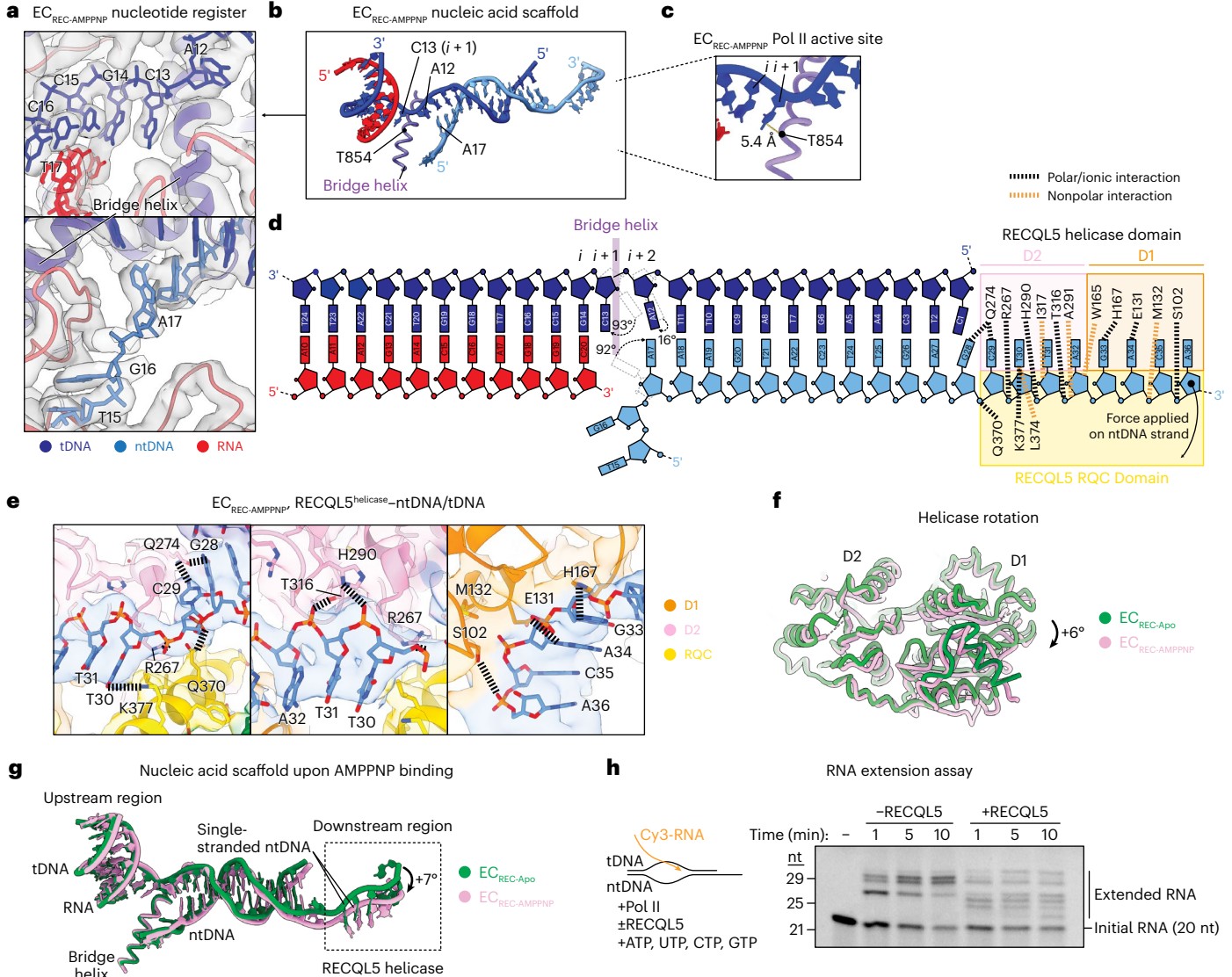

**Fig. 4 | Helicase nucleotide binding induces a post-translocation Pol II conformation. a**, Regions near the Pol II active site upstream (top) and downstream (bottom) of the bridge helix in EC$_{REC-AMPPNP}$. The full cryo-EM map for EC$_{REC-AMPPNP}$ is shown in Extended Data Fig. 7c. Cryo-EM density is shown as a transparent gray surface, with the fitted atomic model in ribbon representation and the DNA bases in stick representation. **b**, Nucleic acid scaffold and Pol II bridge helix in EC$_{REC-AMPPNP}$. **c**, Close-up view of the EC$_{REC-AMPPNP}$ Pol II active site, highlighting the distance between the $i + 1$ site nucleotide base and the RPB1 T854 residue in the bridge helix. **d**, Schematic depicting the nucleic acid scaffold in EC$_{REC-AMPPNP}$, highlighting resolved nucleotides, the position of the bridge helix and residues in the RECQL5 helicase domain that interact with the nucleic acids. Salt bridges and polar interactions are indicated in black and nonpolar interactions are indicated in orange. **e**, Views of the downstream region in EC$_{REC-AMPPNP}$, highlighting molecular interactions between the RECQL5 helicase domain and the ntDNA. The cryo-EM map is shown as a transparent surface, with the fitted atomic model in ribbon representation and the residue side chains and DNA bases in stick representation. Salt bridges and polar interactions are indicated with black dashed lines. Residues of interest are explicitly labeled. **f**, Comparison of the RECQL5 helicase conformation in EC$_{REC-Apo}$ (green) versus EC$_{REC-AMPPNP}$ (pink). The two structures are aligned on the helicase D2 subdomain, revealing a +6° relative rotation of the D1 subdomain. **g**, Superposition of the nucleic acid scaffolds from the EC$_{REC-Apo}$ (green) and EC$_{REC-AMPPNP}$ (pink) structures. **h**, RNA extension assay testing Pol II activity on the nucleic acid scaffold in Fig. 1b in the presence or absence of RECQL5. Left, assay schematic. Image showing Cy3 fluorescence of reaction products separated by denaturing PAGE. Note that RNAs run slightly larger than expected relative to the ladder because of the presence of the Cy3 fluorescent label (initial RNA, 20 nt). Data are representative of three independent experiments.

the post-translocation state (Extended Data Fig. 7g,h), as we observed in EC$_{REC-AMPPNP}$ (Fig. 4c). This result suggests that ATP hydrolysis and phosphate release by RECQL5 do not provide the energy associated with changes in the Pol II translocation configuration; rather, the conformational change of RECQL5 upon ATP binding likely does.

To support our structural results, we additionally sought to evaluate how RECQL5 affects Pol II RNA extension. We first confirmed that Pol II is able to transcribe using the same nucleic acid scaffold used for our structural studies. In the absence of RECQL5, Pol II indeed

efficiently extended the RNA, producing major products corresponding to 28–29 nt (Fig. 4h). Addition of RECQL5 inhibited RNA extension on our scaffold, as expected. This is shown by the formation of smaller products (22–25 nt) not observed in the absence of RECQL5, consistent with RECQL5 serving as a roadblock for Pol II progression, as well as a large decrease in the formation of the longer 28–29-nt transcripts. Interestingly, we also observed a longer 30-nt band above the major 28–29-nt products (Fig. 4h). Notably, the ratio between this 30-nt product and the 28–29-nt products was greatly increased by the presence

**a** Mechanism by which RECQL5 slows Pol II elongation

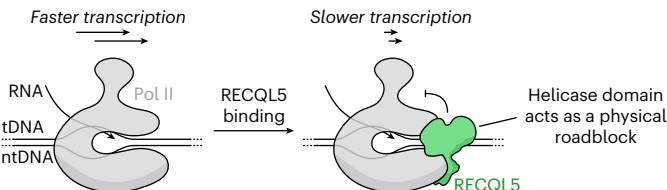

**b** Proposed mechanism by which RECQL5 decreases transcription stress

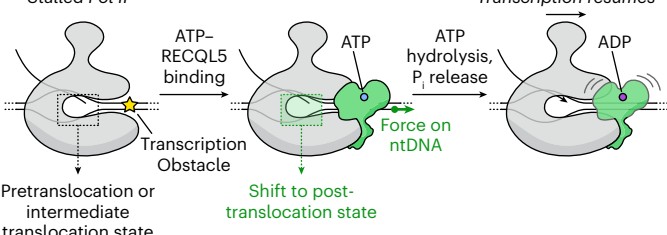

**Fig. 5 | Proposed mechanisms of transcriptional regulation by RECQL5.**
**a**, Schematic depicting the function of RECQL5 as a negative elongation factor. RECQL5 globally slows Pol II transcription, presumably by binding to downstream DNA and acting as a roadblock for Pol II advancement. **b**, Schematic of a second possible function of the RECQL5 helicase domain. During Pol II stalling, RECQL5 may help to reset the translocation status of Pol II through its interactions with the downstream DNA. Initial binding of ATP–RECQL5 to the EC is accompanied by a pulling force on the ntDNA. This mechanical stress is propagated along the DNA to the Pol II active site, shifting it to a post-translocation state that is ready to incorporate a new incoming nucleotide into the growing RNA. Subsequent ATP hydrolysis and phosphate ($P_i$) release by the RECQL5 helicase domain increases its conformational flexibility, which may facilitate transcription restart and bypassing of the helicase roadblock. We propose that this mechanism may help explain how RECQL5 decreases transcription stress in cells.

of RECQL5, with the 29-nt band in particular mostly disappearing. In this assay, RECQL5 is expected to be nucleotide bound because ATP is provided for RNA extension. This observation indicates that nucleotide-bound RECQL5 can promote the conversion of the 28–29-nt products to the longer 30-nt product. Altogether, our structural and biochemical results suggest that RECQL5 has the ability to promote Pol II RNA extension, despite its overall inhibitory effect on Pol II activity.

## Discussion

RECQL5 is a regulator of transcription elongation known to interact directly with Pol II, thereby having important roles in suppressing genome instability. Nevertheless, detailed structural and mechanistic insights into its molecular interplay with Pol II are currently lacking. In this study, we solved several cryo-EM structures of Pol II ECs complexed with RECQL5 in different nucleotide-binding states of the helicase. Our structures reveal the molecular interactions that stabilize RECQL5 binding to the Pol II EC. The RECQL5 IRI module, encompassing the αN helix and KIX domain, wraps around the surface of the RPB1 lower jaw with exquisite molecular complementarity, serving as an anchor point of RECQL5 on Pol II. Meanwhile, the helicase domain binds to the RPB1 upper jaw and downstream DNA. Our structural results suggest that the RECQL5 helicase domain may exert its regulatory effects on transcription through two distinct mechanisms.

Firstly, the helicase domain binds to DNA downstream of the transcribing Pol II, thus likely acting as a steric roadblock that slows transcription, as previously proposed[22] (Fig. 5a). The resolution of our cryo-EM structures allowed us to map critical molecular interactions of the RECQL5 helicase domain with the single-stranded DNA and with the upper jaw of Pol II. We also observed extensive conformational flexibility for the helicase domain in all of our datasets, most pronounced for $EC_{REC-ADP}$, while the position of the IRI module remained unaltered

across the $EC_{REC-Apo}$, $EC_{REC-AMPPNP}$ and $EC_{REC-ADP}$ structures. This result indicates that the IRI–RPB1$^{lower\,jaw}$ interaction is stable and anchors the flexible and more loosely engaged RECQL5 helicase domain to the Pol II EC. Consistent with this, two independent studies[35,36] were published concurrently with our study, also showing that the RECQL5 IRI module binds the RPB1 lower jaw in a highly similar manner as described here, although the RECQL5 helicase domain was not observed. The IRI domain's anchoring role may be bolstered by the interaction between the RECQL5 SRI domain (not included in our truncated RECQL5 construct) and the disordered Pol II C-terminal domain, which may help increase the local concentration of RECQL5 around elongating Pol II (ref. 19). Overall, a helicase roadblock mechanism agrees with observations that RECQL5 can inhibit transcription in vitro (Fig. 4h)[21,22] and globally slow elongation rates in cells[23]. As proposed previously[22], this inhibitory function could prevent Pol II from advancing into areas undergoing DNA repair, given RECQL5's interaction with the MRN complex[37], a double-stranded break sensor.

Secondly, our findings show that RECQL5 has the ability to modulate Pol II's translocation status. In $EC_{REC-Apo}$, the RECQL5 helicase domain appears to push the downstream DNA against Pol II, bending it by −17° and partially unwinding it. Upon AMPPNP binding in $EC_{REC-AMPPNP}$, the RECQL5 helicase domain undergoes an internal conformational change (D2 subdomain rotates +6°) that reorganizes its contacts with the single-stranded ntDNA and pulls it away from Pol II, partially rewinding the DNA. These observations are consistent with the known property of dsDNA to exhibit negative coupling between twisting and stretching under small distortions, overwinding when stretched and underwinding when contracted[38]. This twisting–stretching inverse coupling is of particular importance for DNA-binding proteins that exploit this property by causing local distortions of the B-DNA geometry upon binding[38]. Our structures show that these pushing and pulling actions induced by RECQL5 in different nucleotide-binding states apply torques on the DNA that propagate upstream to the Pol II active site. Crucially, in the case of AMPPNP-bound RECQL5, this mechanism of action induces Pol II to adopt a post-translocation state.

Notably, a previous study showed that RECQL5 not only slows transcription elongation but also reduces transcription stress, which is linked to genome instability[23]. Our finding that RECQL5 can modulate Pol II's translocation state suggests that RECQL5 may act mechanically on Pol II in the cell. Coupled with our observation that RECQL5 can promote longer RNA extension, this raises the possibility that RECQL5 might help restart stalled or paused Pol II transcription (Fig. 5b). In this model, ATP-complexed RECQL5 binds the Pol II EC, upon which its helicase domain pulls outward on the downstream DNA, generating torsion on the DNA that is allosterically transmitted toward the Pol II active site. This stabilizes the post-translocation conformation, which becomes ready to incorporate a new incoming nucleotide into the growing RNA. ATP hydrolysis and phosphate release by the helicase domain do not alter the Pol II translocation state. Instead, they appear to destabilize the binding of the helicase domain to the EC, which we speculate could facilitate Pol II's forward progress. We propose that the effect is to facilitate the restarting of transcription in the event of Pol II pausing, thereby decreasing transcription stress and associated genome instability. This potential model would resolve the apparent contradiction of how a factor that acts as a roadblock can nonetheless favor transcription elongation by lowering the levels of Pol II stalling and backtracking. However, it will be necessary for future studies to directly test this hypothesis by investigating RECQL5's effect on single-nucleotide incorporation using rapid kinetics experiments or optical tweezer approaches. In addition, detailed investigation of how *RECQL5* mutations affect transcription stress and Pol II occupancy in cells will be needed to better evaluate the role of RECQL5 in modulating Pol II's translocation state.

An important question relates to the biological contexts in which RECQL5 contacts Pol II. We used a nucleic acid scaffold with

a single-stranded ntDNA downstream of Pol II to stabilize RECQL5 binding. Such a situation could be encountered in cells because of the generation of R-loops by transcription, in which the nontemplate strand is displaced as single-stranded DNA by the RNA product[39]. RECQL5 could bind to such regions formed by one copy of Pol II and, therefore, come into contact with an upstream copy of Pol II transcribing the same gene. Similarly, loading of RECQL5 at double-stranded breaks through its interaction with the MRN complex[37], followed by its translocation away from the DNA damage site, could also bring it to Pol II. On the other hand, more recent evidence has underscored the critical role of RECQL5 in the resolution of TRCs. RECQL5 helicase activity disassembles RAD51 filaments from the stalled replication fork, an important step in replication restart[13–15]. In this context, it is tempting to speculate that RECQL5's interaction with Pol II could help bring it to TRC site; alternatively, the stalled replication fork could serve as a platform to load RECQL5 so that it can contact and restart Pol II. Therefore, an important outstanding question is whether RECQL5's role in transcriptional regulation is connected to its function in resolving TRCs. RECQL5's unstructured sequence is known to contain a RAD51-interacting region[11] but structural insights are lacking into both this interaction and the process of filament disassembly. Understanding the molecular basis for RECQL5's function in processing TRCs will be an important goal for future studies.

## Online content

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

## Methods

### Protein purification

GST-tagged RECQL5$_{1-620}$ was expressed in *E. coli* BL21-CodonPlus (DE3)-RIL cells (Stratagene) for 16 h at 18 °C and the cells were lysed with a cell disrupter (Avestin)[22]. The clarified lysate was loaded onto a glutathione Sepharose 4 fast-flow column equilibrated in RECQL5 buffer (20 mM Tris pH 8.0, 150 mM NaCl, 10% glycerol and 1 mM dithiothreitol (DTT)) and eluted with a linear gradient to 20 mM glutathione. The GST tag was cut with PreScission protease during dialysis for 14 h. The protein was subsequently purified using HiTrap Heparin HP and Superdex 200 columns (Cytiva) in RECQL5 buffer.

Human Pol II was purified according to a published protocol[40]. HeLa nuclei (114-L culture) was ground under liquid nitrogen using a mortar and pestle and then slowly resuspended in buffer A (cold 50 mM Tris-HCl pH 7.9, 5 mM MgCl$_2$, 0.5 mM ethylenediaminetetraacetic acid (EDTA), 25% glycerol, 5 mM DTT, 1 mM sodium metabisulfite and 1 mM phenylmethylsulfonyl fluoride (PMSF), supplemented with complete EDTA-free protease inhibitor cocktail (Roche)). After sonicating the resuspension for 2 min with stirring, (NH$_4$)$_2$SO$_4$ was added to a final concentration of 0.3 M. The mixture was further sonicated to reduce viscosity and then clarified by centrifugation (125,000$g$, 90 min, 4 °C, Ti45 rotor). The supernatant was adjusted to the conductivity of 0.1 M (NH$_4$)$_2$SO$_4$ buffer through slow addition of buffer A. Then, a 42% (NH$_4$)$_2$SO$_4$ cut was used to precipitate Pol II, followed by centrifugation (70,400$g$, 30 min, 4 °C, Ti45 rotor). The precipitate was resuspended in buffer B (cold 50 mM Tris-HCl pH 7.9, 0.1 mM EDTA, 25% glycerol, 2 mM DTT and 0.1 mM PMSF) with the concentration of (NH$_4$)$_2$SO$_4$ adjusted to 0.15 M. The sample was applied to a DEAE52 column, which was washed with three column volumes of buffer B with 0.15 M (NH$_4$)$_2$SO$_4$ before elution with buffer B with 0.4 M (NH$_4$)$_2$SO$_4$. Protein-containing fractions were pooled and adjusted by dialysis to the conductivity of 0.2 M (NH$_4$)$_2$SO$_4$ buffer, supplemented with 0.1% NP-40 substitute and immunoprecipitated overnight at 4 °C using anti-RPB1 antibody (clone 8WG16, Biolegend) crosslinked to protein G Sepharose fast-flow resin (Cytiva). The resin was washed three times with buffer C (cold 25 mM HEPES pH 7.9, 0.2 mM EDTA, 10% glycerol, 2 mM DTT, 0.1 mM PMSF and 0.05% NP-40 substitute) with 0.5 M (NH$_4$)$_2$SO$_4$, followed by two washes with buffer C with 0.2 M (NH$_4$)$_2$SO$_4$. Pol II was eluted through four sequential incubations with buffer C supplemented with 0.23 M (NH$_4$)$_2$SO$_4$ and 1 mg ml$^{-1}$ RPB1 triheptapeptide repeat (sequence: (YSPTSPS)$_3$). Concentrated eluate was flash-frozen in liquid N$_2$. An SDS–PAGE gel of purified proteins is shown in Extended Data Fig. 7b.

### Preparation of EC$_{REC-Apo}$ complex

For purification of EC$_{REC-Apo}$ (Pol II bound to the nucleic acid scaffold and RECQL5$_{1-620}$-D157A with no nucleotide), Pol II was incubated first with a tenfold molar excess of the nucleic acid scaffold and then with a tenfold molar excess of RECQL5 while immobilized to the anti-RPB1–protein G resin during the Pol II purification procedure described above[22]. The resin was washed and the EC$_{REC-Apo}$ complex was eluted with RPB1 triheptapeptide repeat. Then, the complex was diluted with transcription buffer (20 mM HEPES pH 8.0, 4 mM MgCl$_2$, 50 mM KCl, 0.05% NP-40 substitute and 1 mM Tris-(2-carboxyethyl)phosphine (TCEP)) and crosslinked with 0.02% glutaraldehyde for 10 min, followed by quenching with 100 mM Tris. Complex was then aliquoted and flash-frozen with liquid N$_2$.

### Preparation of EC$_{REC-AMPPNP}$ and EC$_{REC-ADP}$ complexes

Nucleotide-bound complexes were assembled and purified using a pulldown strategy (Extended Data Fig. 7a). First, tDNA (5′-CTCAAGTACTTACGCCTGGTCATTACTA-3′) and RNA (5′-UAUAUGCAUAAAGACCAGGC-3′) were annealed by incubating at 90 °C for 5 min and then cooling to 4 °C at a rate of 0.2 °C s$^{-1}$. Pol II (diluted from 513 nM stock)

was mixed with the tDNA–RNA hybrid and incubated at room temperature for 20 min. Then, desthiobiotinylated ntDNA (5′-/5deSBioTEG/TAGTAAACTAGTATTGAAAGTACTTGAGCTTAGACAGCATGTC-3′) was added and the mixture was incubated at room temperature for 20 min. All oligonucleotides were purchased from Integrated DNA Technologies. Finally, RECQL5$_{1-620}$ and nucleotide were added and the mixture was incubated at room temperature for 20 min. Because D157 stabilizes bound ADP through water-mediated interactions with the coordinated Mg$^{2+}$ ion[34], we used wild-type RECQL5 for these studies instead of the D157A mutant. The final mixture contained 350 nM Pol II, 350 nM nucleic acid scaffold, 7 µM RECQL5 and 1 mM AMPPNP or ADP, diluted in transcription buffer. After the incubation, Dynabeads MyOne Streptavidin T1 was added (6.25 µl of beads per 12.5 µl of input) and the mixture was incubated at room temperature for 75 min. The beads were washed twice with transcription buffer containing 1 mM of AMPPNP or ADP, as appropriate. Then, the complex was eluted by incubating the beads twice with elution buffer (20 mM HEPES pH 8.0, 4 mM MgCl$_2$, 50 mM KCl, 0.05% NP-40 substitute, 1 mM TCEP, 5 mM biotin, 3% trehalose and 1 mM AMPPNP or ADP) at 37 °C for 15 min. The eluted complex was crosslinked with 0.02% glutaraldehyde at room temperature for 10 min and quenched by incubating with 100 mM Tris-HCl pH 8.0 at room temperature for 5 min. The crosslinked complexes were used to prepare cryo-EM grids on the same day.

### Cryo-EM sample preparation

Cryo-EM specimens were deposited on graphene oxide (GO)-coated[41] Quantifoil grids (1.2/1.3 300-mesh, carbon on gold). Grids were cleaned with chloroform, glow-discharged using a Tergeo-EM plasma cleaner (PIE Scientific), incubated for 2 min with 1 mg ml$^{-1}$ polyethylenimine (Polysciences) in 25 mM HEPES pH 7.9, washed twice with H$_2$O and air-dried. Then, grids were incubated for 2 min with 0.2 mg ml$^{-1}$ GO stock solution, washed twice with H$_2$O and air-dried. To prepare the GO stock solution, we diluted GO in 1:2 methanol and H$_2$O (v/v), sonicated the mixture, centrifuged at 4,000$g$ for 10 min (to remove small GO sheets), resuspended the pellet in 1:2 methanol and H$_2$O (v/v), further sonicated the mixture and finally collected the supernatant after centrifugation at 1,000$g$ for 1 min (to remove GO aggregates). We found that a 1:2 methanol and H$_2$O (v/v) solution facilitated the deposition of a continuous GO layer on the grid. Grids were either used on the same day or saved and gently glow-discharged before use. Onto each grid, 3.5 µl of sample was deposited, followed by incubation for 30 s at 22 °C with 100% humidity in a Vitrobot Mark IV (Thermo Fisher Scientific). Then, the grid was blotted for 10 s with a blot force of 10 and vitrified by plunging into liquid ethane with a liquid N$_2$ bath.

### Cryo-EM data collection

All cryo-EM data were acquired as dose-fractionated videos with a 300-kV Titan Krios G3 cryo-EM instrument (Thermo Fisher Scientific) using a K3 direct electron detector (Gatan). A total exposure dose of 50 e$^-$ per Å$^2$ fractionated across 50 frames was used during video frame recording, with defocus values ranging from approximately −0.8 to −1.8 µm. All data collection processes were automatically controlled using SerialEM[42] and parameters are summarized in Table 1.

For the EC$_{REC-Apo}$ sample, 6,190 videos (dataset 1, EMPIAR-12711) were collected in super-resolution counting mode at ×81,000 magnification using correlated double sampling (CDS) and a super-resolution pixel size of 0.525 Å per pixel. From dataset 1, the EC$_{REC-Apo}$, EC$_{Free}$ and EC$_{REC-Apo (IRI-focused)}$ structures were produced as described below.

For the EC$_{REC-AMPPNP}$ and EC$_{REC-ADP}$ complexes, data were acquired in the same instrument as described above but using non-CDS and non-super-resolution mode to increase throughput at ×81,000 magnification with a physical pixel size of 1.048 Å per pixel. For EC$_{REC-AMPPNP}$ and EC$_{REC-ADP}$, a total of 12,100 videos (dataset 2, EMPIAR-12721) and 9,048 videos (dataset 3, EMPIAR-12722) were collected, respectively.

## Cryo-EM image processing

Data processing of all images was conducted using cryoSPARC (version 4.5.3)[43,44] and RELION (version 5)[45,46] software, as detailed in Extended Data Figs. 1, 3, 5, 8 and 9. For simplicity, we describe in detail the data analysis workflow followed for dataset 1 ($EC_{Free}$, $EC_{REC-Apo}$ and $EC_{REC-Apo (IRI-focused)}$ structures). We note that the analyses for datasets 2 and 3 ($EC_{REC-AMPPNP}$ and $EC_{REC-ADP}$ structures, respectively) were performed following similar workflows as for $EC_{REC-Apo}$.

**Initial dataset 1 processing.** For dataset 1, the 6,190 video frames collected were aligned using patch motion correction within cryoSPARC[43,44]. Then, defocus estimation and contrast transfer function (CTF) fitting were performed using patch CTF estimation. In the corrected micrographs, we could readily observe particles with the size and features expected for Pol II ECs (Extended Data Fig. 2a).

A preliminary round of data processing was performed on 100 randomly selected micrographs, where particles were picked using the blob picker algorithm. Later, multiple rounds of two-dimensional (2D) classification and particle selection cycles were carried out to obtain suitable 2D templates for the following template picker job on all micrographs. Three subsequent rounds of 2D classification and particle selection cycles resulted in a set of 2,240,800 particles, from which 300,000 particles were randomly selected to perform an ab initio 3D reconstruction ($n = 3$). Two of three 3D ab initio maps were used as references to run a heterogeneous refinement using the full particle set ($n = 4$). From these classes, one particular class, containing 39.1% of the population (875,262 particles), showed defined structural features, while the other classes displayed broken complexes and/or poor low-resolution reconstructions. The particles corresponding to the best class were re-extracted using a box size of 320 pixels × 320 pixels (without binning), resulting in 871,524 particles (duplicate particles removed), and further subjected to a homogeneous refinement job, obtaining a 3D reconstruction at 2.4-Å overall resolution (Fourier shell correlation (FSC) = 0.143). At low-threshold levels, two fuzzy regions appeared next to the EC, resembling the positioning of the RECQL5 helicase (region A) and KIX domains (region B) observed in the low-resolution cryo-EM structure of this complex reported previously by our group[22] (Extended Data Fig. 3). These ill-defined densities suggested a large degree of local heterogeneity or partial occupancy of RECQL5, which was quite difficult to sort out by standard classification methods. Therefore, we implemented the data analysis pipeline detailed below.

**$EC_{REC-Apo}$ processing.** The 871,524 particles in cryoSPARC were exported to RELION[45,46] and subjected to 3D refinement. The RECQL5 helicase domain appeared notably less stable than the KIX domain. Therefore, we aimed first to resolve the local heterogeneity in the helicase domain (region A).

Using the volume segmentation tool in ChimeraX[47,48] and the mask creation job in RELION, we generated a binary mask involving the region assigned to the RECQL5 helicase domain. We then performed a particle subtraction job to keep the signal inside the mask, while simultaneously recentering the subtracted particles on the mask and reboxing them to a box size of 180 pixels × 180 pixels. Then, using the relion_reconstruct program, we backprojected the subtracted particles to generate a low-pass-filtered 3D reconstruction to be used as a 3D reference for the next 3D classification job. This 3D classification job was performed without alignment, applying a contoured mask, generating four ($n = 4$) classes and using a $T$ value of 15 and blush regularization. One of the four classes, containing 23.7% of the population (206,338 particles) and displaying better defined features, was selected and subjected to subtraction reversion to recover the full particle information. The reverted particles were then backprojected to generate a new 3D reference and then subjected to 3D refinement resulting in a reconstruction at 3.3-Å overall resolution. In this map, the RECQL5 helicase domain showed substantial improvement (interestingly, the KIX domain density also improved) and defined structural features started to become apparent. We followed up with an additional round of this strategy; however, in the second round of 3D classification without alignment, the $T$ value was increased to 500. We suspected that a larger $T$ value would be helpful because more relative weight would be considered on the actual experimental data (particles) along the classification cycles. One major class, accounting for 42.1% of the population (86,831 particles) and displaying clear secondary structure features, was selected and subjected to subtraction reversion, backprojection and 3D refinement (same as in the first cycle). The resulting reconstruction displayed a well-defined RECQL5, although some fuzziness was still observed for the helicase D1 subdomain (orientation in Fig. 1a). Therefore, we performed two additional rounds of this particle subtraction, 3D classification, subtraction reversion and 3D refinement cycle to improve the helicase D1 subdomain region. To this aim, different combinations of particle reboxing sizes and $T$ values were used because of the smaller region under analysis. Ultimately, 24,323 particles were used to obtain the final cryo-EM reconstruction of $EC_{REC-Apo}$ at 3.2-Å overall resolution (FSC = 0.143) (Extended Data Fig. 2c). In this map, the Pol II EC core had the highest local resolution, while the fully visible RECQL5 helicase and KIX domain regions had local resolutions ranging between 3.7 Å and 5.9 Å. The final cryo-EM reconstruction was postprocessed using the DeepEMhancer sharpening program[49].

The data analysis approach described above enabled us to improve the helicase domain resolution within the context of the full complex and was also used for $EC_{REC-AMPPNP}$ and $EC_{REC-ADP}$. For the purpose of elucidating molecular interactions between RECQL5 and the Pol II EC, this workflow worked better than standard approaches such as focused classification and focused refinement only, which only resulted in an improved helicase domain map isolated from the rest of the EC. By contrast, our approach allowed us to map the full $RECQL5_{1-620}$ construct and describe its interactions with different regions of the Pol II EC (Fig. 2).

**$EC_{REC-Apo (IRI-focused)}$ processing.** We used a similar workflow to further improve the local resolution of the RECQL5 KIX domain region (region B) (Extended Data Fig. 5). Starting over from the 871,524 particles exported to RELION, we subjected the particles to 3D refinement and then adapted our previous data analysis approach to solve the local heterogeneity in this area. By rigid-body fitting both initial Pol II EC coordinates[27] (PDB 5FLM) and a RECQL5 model predicted by AlphaFold 3 (ref. 50) into the $EC_{REC-Apo}$ cryo-EM map, we observed that the RECQL5 IRI module (harboring the αN helix and KIX domain) is positioned to interact with the lower jaw of the Pol II RPB1 subunit. Therefore, we generated a binary mask involving both the RPB1 lower jaw and the RECQL5 IRI module regions. We then performed a particle subtraction job to keep the signal inside the mask, recenter the subtracted particles on the mask and rebox them to a box size of 180 pixels × 180 pixels. Then, we backprojected the subtracted particles to generate its own low-pass filtered 3D reconstruction to be used as a 3D reference for the next 3D classification job. This 3D classification job was performed without alignment, applying a contoured mask, generating four ($n = 4$) classes and using a $T$ value equal to 20 together with blush regularization. One of these four classes, harboring 29.0% of the population (253,055 particles), displayed better features clearly observed both in the map and in the slice view representation. This class was selected and subjected to subtraction reversion to recover the full particle information. The reverted particles were then backprojected and subjected to 3D refinement, obtaining a reconstruction at 3.2-Å overall resolution. In this 3D map, a clear improvement of the RECQL5 KIX domain region was observed. Thus, we decided to perform one additional round of this particle subtraction, 3D classification, subtraction reversion and 3D refinement cycle. Ultimately, 103,215 particles were used to obtain the final cryo-EM reconstruction of the $EC_{REC-Apo (IRI-focused)}$ at 2.8 Å overall resolution (FSC = 0.143) (Extended Data Fig. 2d). In this map, the region

of interest involving the RECQL5 KIX domain was fully visible and displayed a local resolution range of about 3.2–4.0 Å, a major improvement compared to the $EC_{REC-Apo}$ structure. This final cryo-EM map was then postprocessed using the DeepEMhancer sharpening program[49].

**$EC_{Free}$ processing.** As mentioned above, the 2.4-Å-resolution cryo-EM map of $EC_{REC-Apo}$ obtained from homogeneous refinement before RELION processing showed two fuzzy regions next to the EC corresponding to the RECQL5 helicase (region A) and KIX domains (region B) (Extended Data Fig. 3). As these ill-defined densities suggested partial occupancy of RECQL5, we performed global 3D classification without alignment in cryoSPARC (Extended Data Fig. 1). From the four 3D classes generated, one class harboring 24.3% of the total population (212,196 particles) showed no density attributable to any RECQL5 domain and was, therefore, recognized as $EC_{Free}$. This class population was selected and subjected to homogeneous refinement, resulting in a 2.6-Å-resolution (FSC = 0.143) cryo-EM map. Then, an additional round of 2D classification was used to remove low-resolution particles and remaining contaminants, resulting in a set of 174,428 particles. Lastly, nonuniform refinement was used to obtain the final $EC_{Free}$ cryo-EM structure at 2.4-Å overall resolution (FSC = 0.143) (Extended Data Fig. 2b). This final cryo-EM map was then postprocessed using the DeepEMhancer sharpening program[49].

**Dataset 2 and dataset 3 processing.** The image processing corresponding to datasets 2 and 3 was performed following the same approach used for dataset 1 to obtain the $EC_{REC-Apo}$ structure. As detailed in Extended Data Figs. 8 and 9, a total of 80,622 and 17,442 particles were used to obtain the final structures of $EC_{REC-AMPPNP}$ (3.2-Å overall resolution, FSC = 0.143) and $EC_{REC-ADP}$ (3.7-Å overall resolution, FSC = 0.143), respectively. Both maps were then postprocessed independently using the DeepEMhancer sharpening program[49].

### Model building, refinement and validation
For the $EC_{Free}$, $EC_{REC-Apo}$, $EC_{REC-AMPPNP}$ and $EC_{REC-ADP}$ structures, the initial coordinates of the EC were obtained by rigid-body fitting the atomic model of the transcribing mammalian Pol II (PDB 5FLM)[27] into the corresponding postprocessed maps using ChimeraX[47,48]. For RECQL5, the initial coordinates were obtained from different sources. For the helicase and RQC domains, the initial model was taken from the X-ray structure of the human RECQL5 in apo form (PDB 5LB8)[34], whereas, for the αN helix and KIX domain, the atomic model was predicted using AlphaFold 3 (ref. 50). These RECQL5 model regions were then semiautomatically docked and rigid-body fitted into the corresponding sharpened map.

For $EC_{REC-Apo\ (IRI-focused)}$, the transcribing mammalian Pol II (PDB 5FLM)[27] model and the predicted coordinates for the RECQL5 IRI module (αN helix and KIX domain) were both rigid-body fitted into the sharpened map. For subsequent model building and refinement, we only kept the coordinates corresponding to the Pol II RPB1 lower jaw (residues 1162–1305) and the RECQL5 IRI module (harboring the αN helix and KIX domain, residues 498–620) because these were the interacting regions of interest for this structure.

For each complex, the models were then iteratively rebuilt in Coot[51] and refined using the real space refinement program in PHENIX[52]. All validation and refinement statistics are shown in Table 1. The overall fit of the models to the maps is shown in Fig. 1e,f and Supplementary Fig. 1.

The FSCs for map versus map were obtained from the half-maps for each structure considering an applied contoured mask. The FSCs for map versus model were obtained by running a validation job in PHENIX of the corresponding final refined atomic model against the unsharpened full map. A custom Python script was used to create FSC plots.

### Structural visualization and interpretation
All the structural comparisons and superpositions, as well as rotation, r.m.s.d. and distance measurements were performed in ChimeraX[47,48].

The difference maps shown in Extended Data Fig. 7e were generated in ChimeraX as follows. First, we created a density map from the coordinates corresponding to Pol II EC and RECQL5 only, without considering the nucleotide bound; then, this map was subtracted from the corresponding full cryo-EM map for $EC_{REC-AMPPNP}$ or $EC_{REC-ADP}$. Overall, all main figures show the sharpened cryo-EM maps and the final refined atomic models unless otherwise specified.

### RNA extension assay
RNA extension assays were conducted similarly to previous studies[22,53]. All oligonucleotides were purchased from Integrated DNA Technologies. First, tDNA (5′-CTCAAGTACTTACGCCTGGTCATTACTA-3′) and Cy3-labeled RNA (5′-/5Cy3/UAUAUGCAUAAAGACCAGGC-3′) were annealed, mixed with Pol II and diluted in assay buffer (20 mM HEPES pH 8.0, 4 mM $MgCl_2$, 100 mM KCl and 1 mM TCEP) and incubated for 20 min at room temperature. ntDNA (5′-TAGTAAACTAGTATTGAAAGTACTTGAGCTTAGACAGCATGTC-3′) was added and the mixture was incubated for 20 min at room temperature, followed by addition of RECQL5 (or an equivalent volume of assay buffer) and another incubation for 20 min at room temperature. Then, nucleoside triphosphates (NTPs) were added to initiate the reaction and the mixture was incubated at 30 °C. The final reactions contained 50 nM Pol II, 200 nM nucleic acid scaffold, 2 μM RECQL5 (if added) and 800 μM each of ATP, uridine triphosphate (UTP), cytidine triphosphate (CTP) and guanosine triphosphate (GTP), all diluted in assay buffer. At appropriate time points, an aliquot of the reaction was removed and quenched with an equal volume of stop buffer (6.4 M urea, 50 mM EDTA in 1× Tris–borate–EDTA (TBE) buffer). Then, proteinase K (New England Biolabs) was added to a final concentration of 0.95 μg μl$^{-1}$ and the mixture was incubated at 30 °C for 15 min. An equal volume of 2× RNA loading dye (New England Biolabs) was added and the samples were heated at 70 °C for 10 min. Samples were subjected to electrophoresis using a 15% TBE–urea gel (Bio-Rad) in 0.5× TBE (150 V, 1.5 h) and imaged on a Typhoon FLA 9500 scanner (Cytiva) detecting Cy3 fluorescence. Raw images were processed by adjusting the image levels to improve the tonal range using Adobe Photoshop 2023. All adjustments were applied to the full image.

### Reporting summary
Further information on research design is available in the Nature Portfolio Reporting Summary linked to this article.

## Data availability
All data pertaining to this paper are provided within the paper or accessible from public repositories. The cryo-EM density maps and their respective atomic coordinate files were deposited to the EM Data Bank and PDB under the following accession codes: EMD-48071 and PDB 9EHZ ($EC_{Free}$), EMD-48073 and PDB 9EI1 ($EC_{REC-Apo}$), EMD-48074 and PDB 9EI2 ($EC_{REC-Apo\ (IRI-focused)}$), EMD-48075 and PDB 9EI3 ($EC_{REC-AMPPNP}$) and EMD-48076 and PDB 9EI4 ($EC_{REC-ADP}$). Raw cryo-EM videos were deposited to the EM Public Image Archive under the following accession codes: EMPIAR-12711 (dataset for $EC_{Free}$, $EC_{REC-Apo}$ and $EC_{REC-Apo\ (IRI-focused)}$), EMPIAR-12721 (dataset for $EC_{REC-AMPPNP}$) and EMPIAR-12722 (dataset for $EC_{REC-ADP}$). In addition to those reported in this work, structures were obtained from the PDB under accession codes 2VUM, 1I6H, 6EXV, 5FLM, 5LB8, 8JH2 and 7NW0. Source data are provided with this paper.

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

## Acknowledgements

We thank the members of the E.N. Lab for important discussions, D. Toso and R. Thakkar at the Cal-Cryo facility at QB3-Berkeley for help with EM data acquisition, F. Burgos-Bravos for biochemical advice, P. Tobias, K. Stine and V. Marquez for computing support and the UC Berkeley Cell Culture Facility for providing HeLa cells. We thank C. Bustamante for his feedback on the manuscript. This work was supported by the National Institutes of Health grant R35 GM127018 to E.N. E.N. is a Howard Hughes Medical Institute investigator. The funders had no role in study design, data collection and analysis, decision to publish or preparation of the manuscript.

## Author contributions

E.N., A.J.F.A. and N.Z.L. conceptualized the study and designed the experiments. A.J.F.A. and N.Z.L. prepared the cryo-EM samples and collected and analyzed the cryo-EM data. A.J.F.A. developed the cryo-EM processing approach. N.Z.L. performed the RNA extension assays. N.Z.L., J.F. and S.A.K. purified the proteins. P.G. assisted with graphene oxide grid fabrication and provided technical advice regarding cryo-EM sample preparation. S.A.K. performed initial sample preparation. B.K. prepared the cryo-EM grids and collected the initial cryo-EM dataset. E.N. supervised the study. A.J.F.A., N.Z.L. and E.N. wrote the manuscript and all authors edited the manuscript.

## Competing interests

The authors declare no competing interests.

## Additional information

**Extended data** is available for this paper at https://doi.org/10.1038/s41594-025-01611-8.

**Correspondence and requests for materials** should be addressed to Eva Nogales.

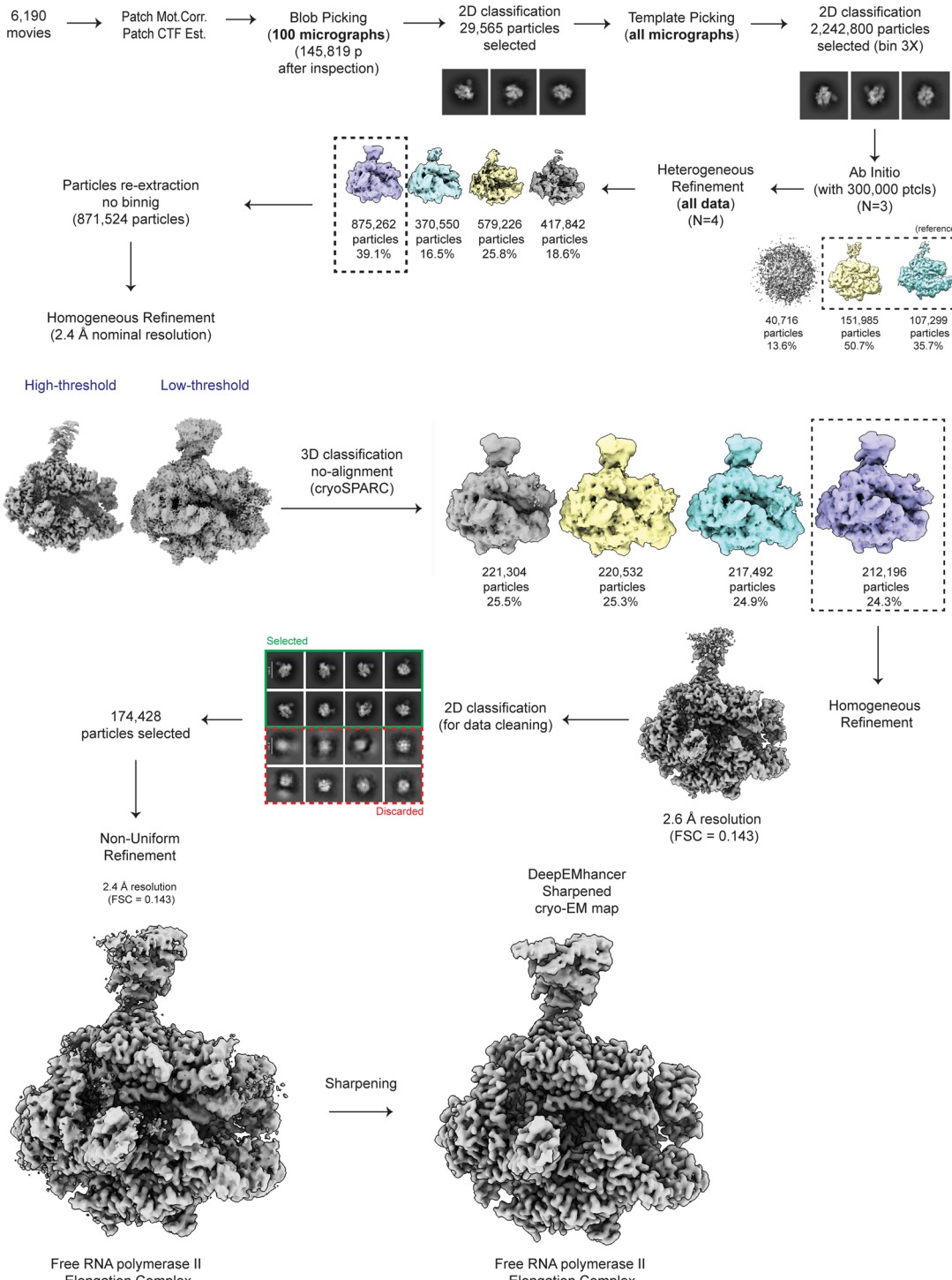

**Extended Data Fig. 1 | Cryo-EM processing workflow for EC_Free.** Workflow showing cryo-EM processing steps for the EC_Free structure.

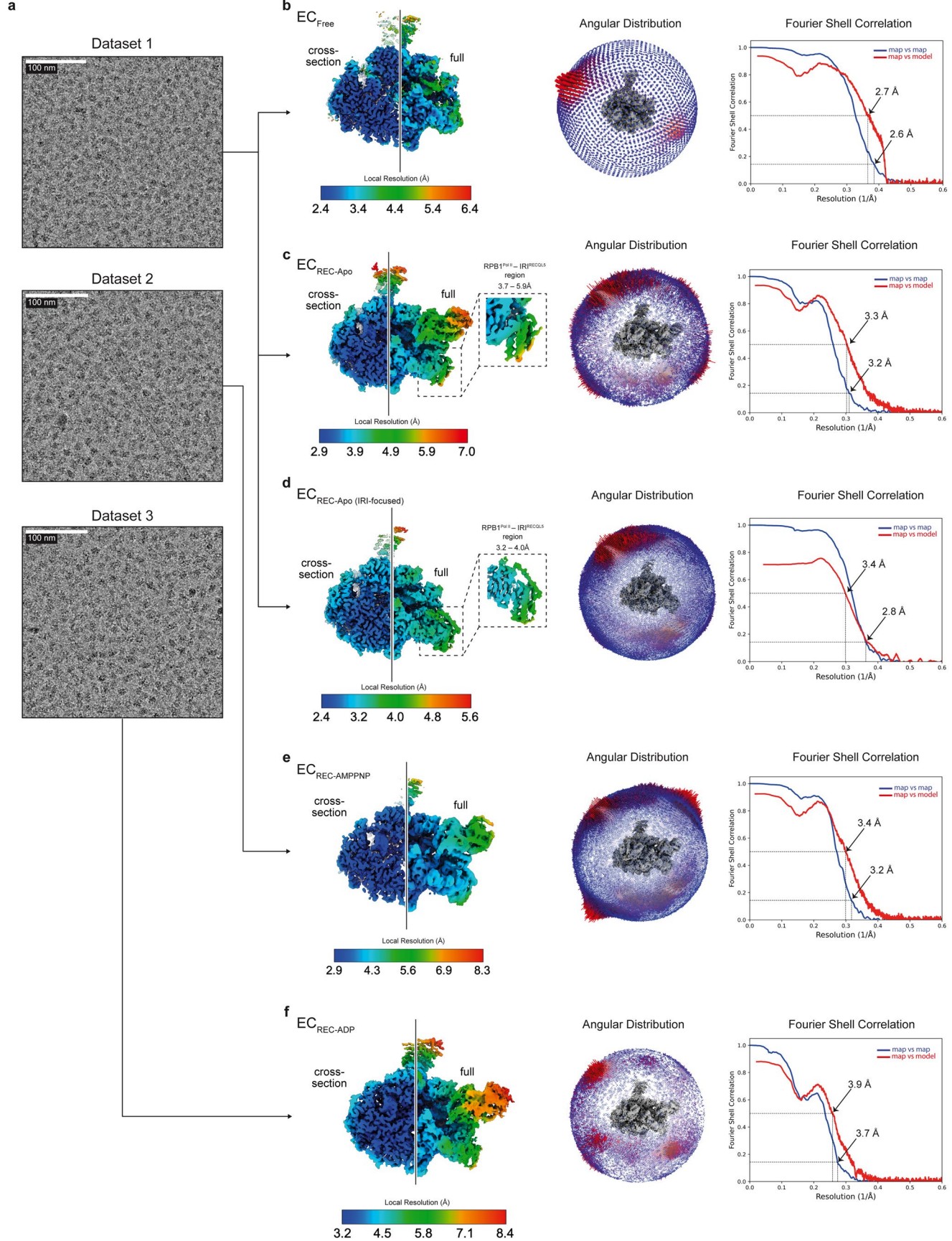

**Extended Data Fig. 2 | See next page for caption.**

**Extended Data Fig. 2 | Cryo-EM data collection and resolution estimation. a**, Examples of cryo-EM corrected micrographs for the datasets analyzed. Scale bars are indicated. **b–f**, Unsharpened cryo-EM map colored by local resolution as indicated in the color key (left), angular distribution plot (middle), and Fourier Shell Correlation (FSC) map vs map and map vs model plots (right) for the EC$_{Free}$ (**b**), EC$_{REC-Apo}$ (**c**), EC$_{REC-Apo (IRI-focused)}$ (**d**), EC$_{REC-AMPPNP}$ (**e**), and EC$_{REC-ADP}$ (**f**) structures. For the local resolution plots, the line separates a cross-section on the left and a view of the cryo-EM map surface on the right. In **c** and **d**, the insets show a close-up view of the RPB1 lower jaw and IRI module colored by local resolution.

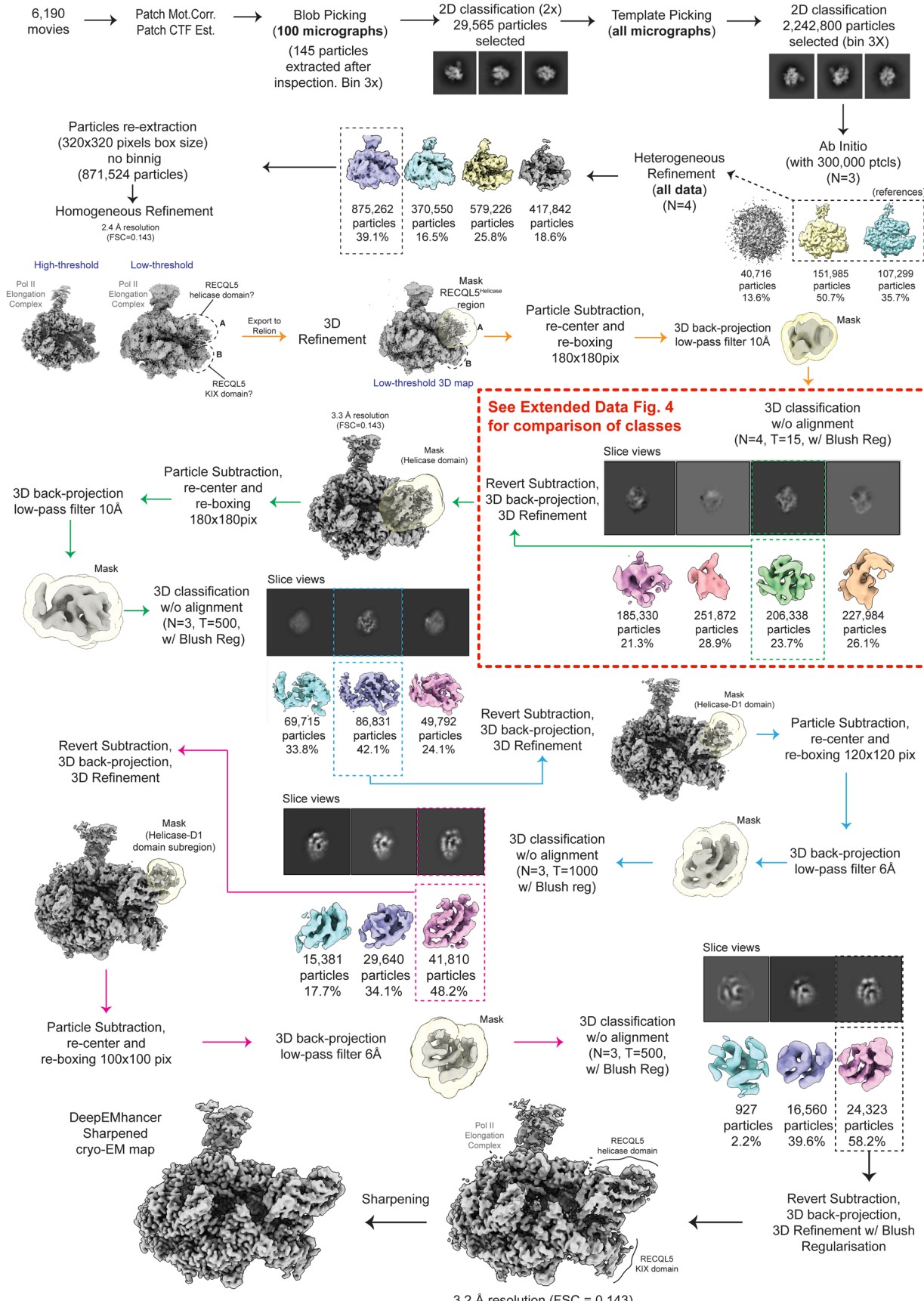

**Extended Data Fig. 3 | Cryo-EM processing workflow for EC_REC-Apo.** Workflow showing cryo-EM processing steps for the EC_REC-Apo structure.

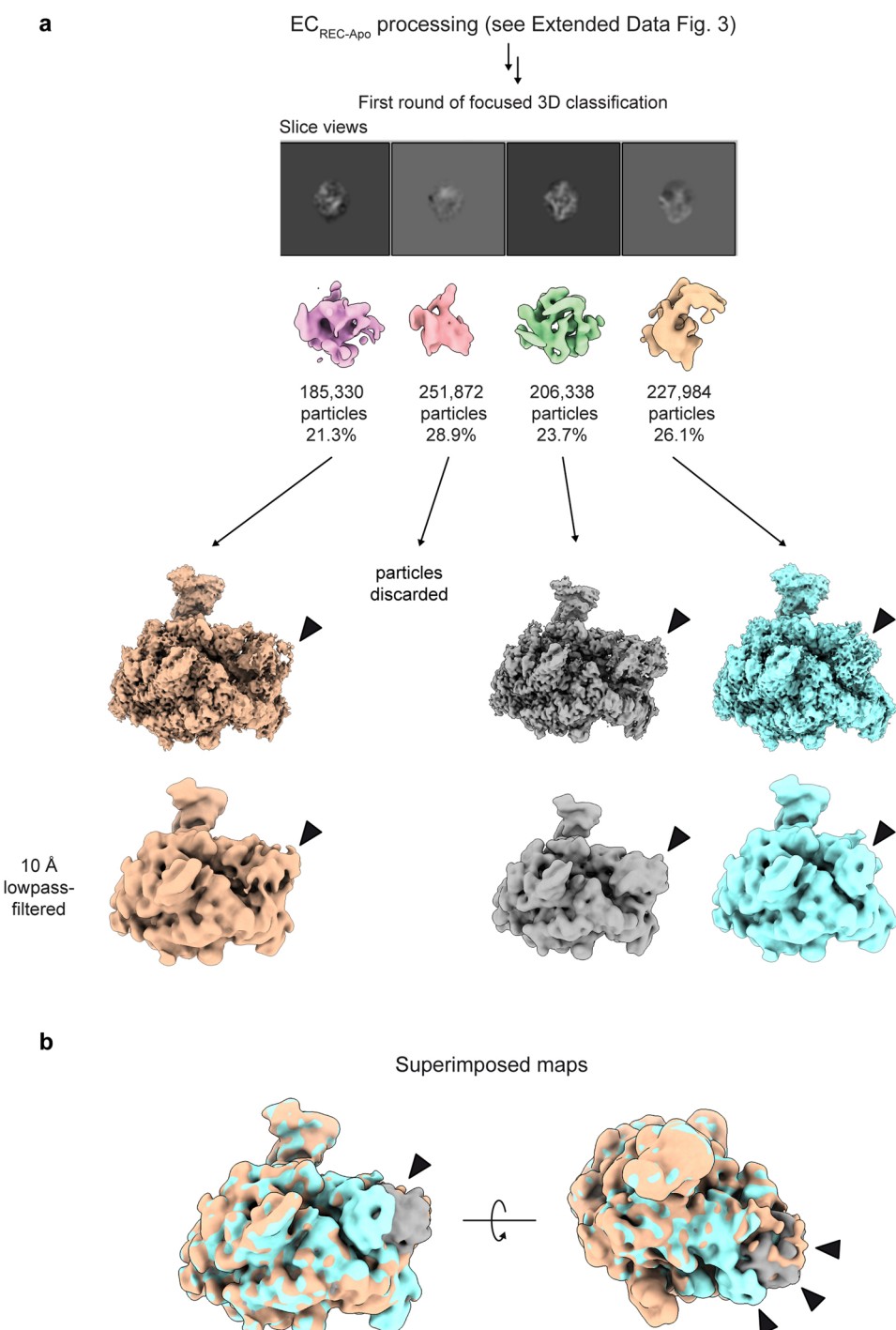

**a** EC$_{REC-Apo}$ processing (see Extended Data Fig. 3)

First round of focused 3D classification

Slice views

185,330 particles 21.3%

251,872 particles 28.9%

206,338 particles 23.7%

227,984 particles 26.1%

particles discarded

10 Å lowpass-filtered

**b** Superimposed maps

**Extended Data Fig. 4 | RECQL5 helicase domain is flexible in the EC$_{REC-Apo}$ dataset. a**, Results from the first focused 3D classification step (see Extended Data Fig. 3). Classification was carried out on subtracted particles, focusing on the helicase domain. After classification, subtraction was reverted and particles were back-projected to obtain 3D reconstructions, which were subsequently refined. In three classes showing strong signal for the helicase domain, which are depicted in orange, gray, and cyan, the helicase domain occupies different positions at the Pol II DNA entry site. The gray map represents the class that was selected for further classification and refinement to obtain the final EC$_{REC-Apo}$ structure. Arrows denote density corresponding to the helicase domain. **b**, Superposition of the three lowpass-filtered maps in **a** with arrows highlighting the different positions the helicase domain adopts.

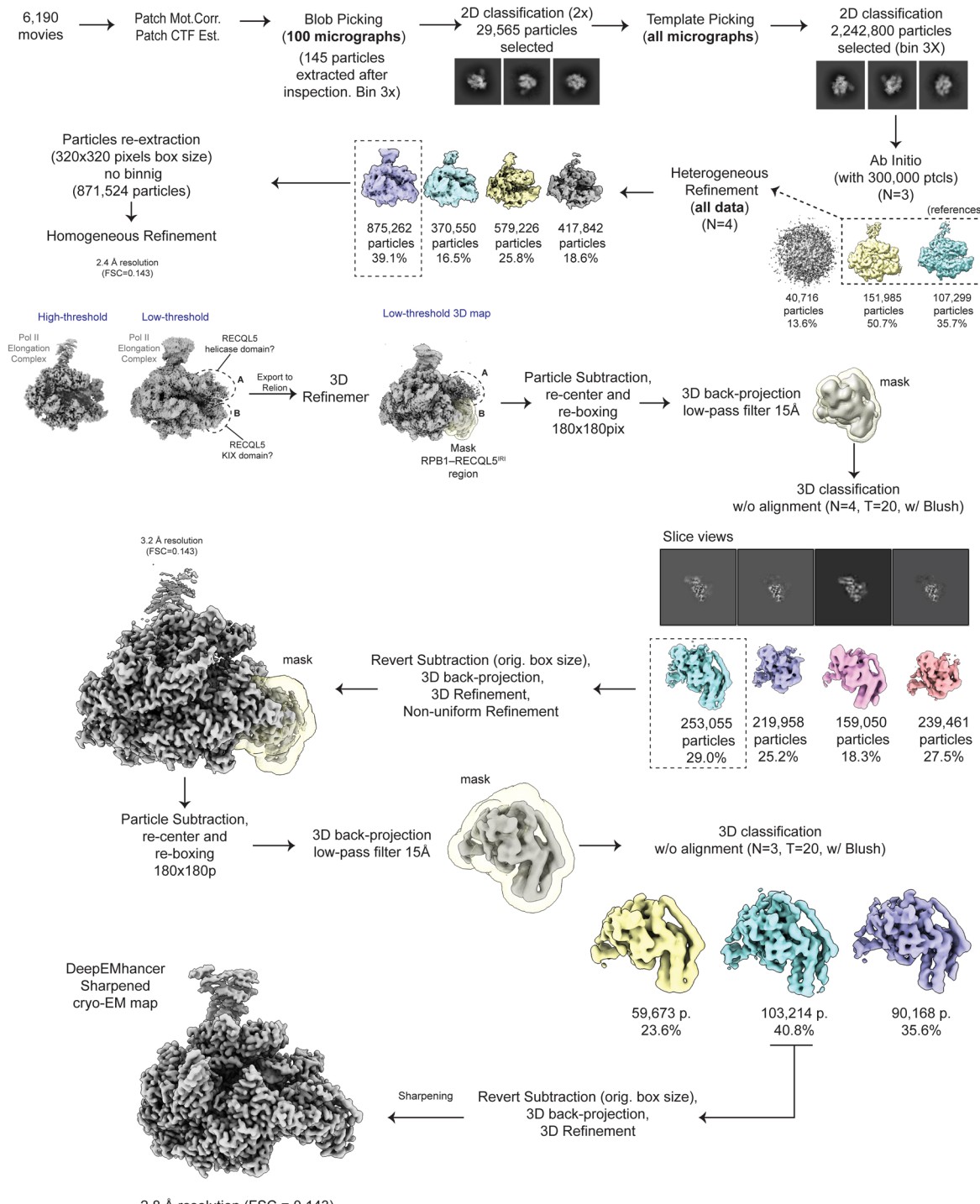

**Extended Data Fig. 5 | Cryo-EM processing workflow for EC_REC-Apo (IRI-focused).** Workflow showing cryo-EM processing steps for the EC_REC-Apo (IRI-focused) structure.

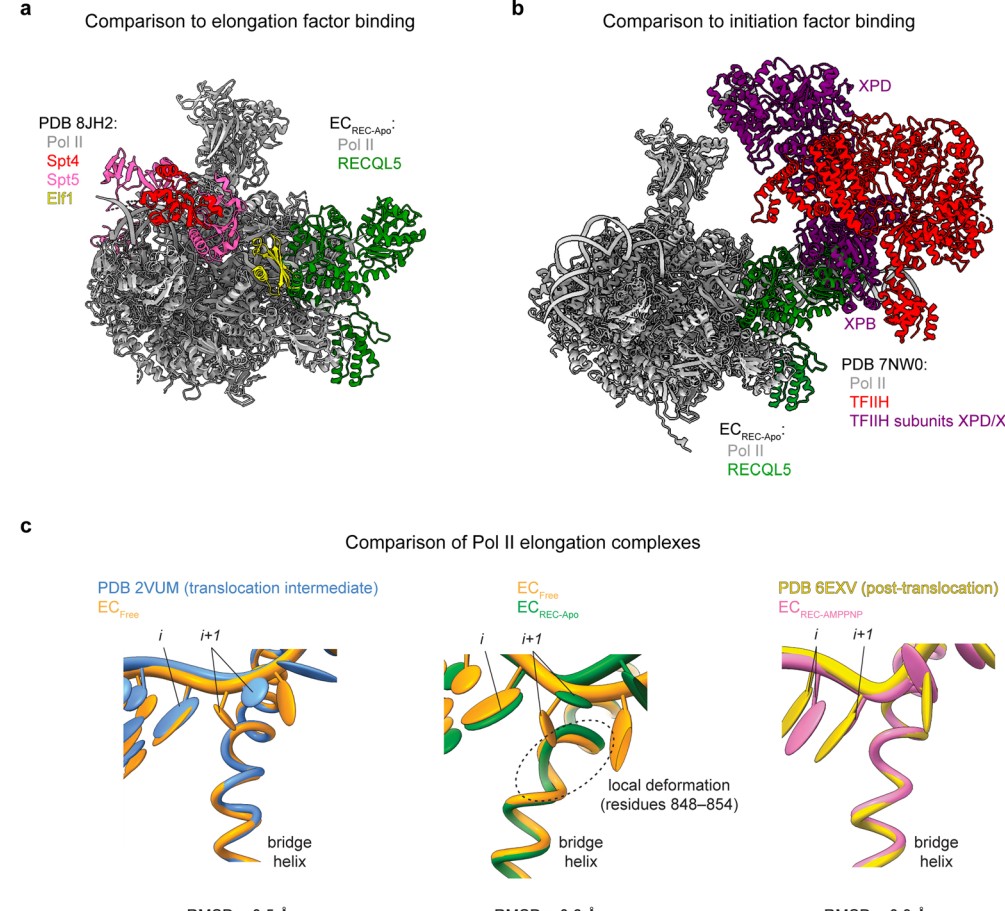

**Extended Data Fig. 6 | Structural comparisons of Pol II complexes.**
**a**, Superposition of EC$_{REC-Apo}$ with PDB$_{8JH2}$ (Pol II–Elf1–Spt4/5–nucleosome structure, nucleosome hidden for clarity). **b**, Superimposition of EC$_{REC-Apo}$ with PDB$_{7NW0}$ (Pol II pre-initiation complex, initiation factors besides TFIIH hidden for clarity). **c**, Superposition of the bridge helix in EC$_{Free}$ vs PDB$_{2VUM}$ (left), EC$_{Free}$ vs EC$_{REC-Apo}$ (middle), and EC$_{REC-AMPPNP}$ vs PDB$_{6EXV}$ (right). RMSDs between the

structures for each comparison are shown below. Visual inspection of the tDNA base in the *i* and *i* + 1 sites across the models (Figs. 3d, e, h and 4c), as well as the RMSDs reported here, indicate that EC$_{Free}$ and EC$_{REC-Apo}$ adopt distinct translocation intermediates, while EC$_{REC-AMPPNP}$ adopts a post-translocation state. Color code is indicated.

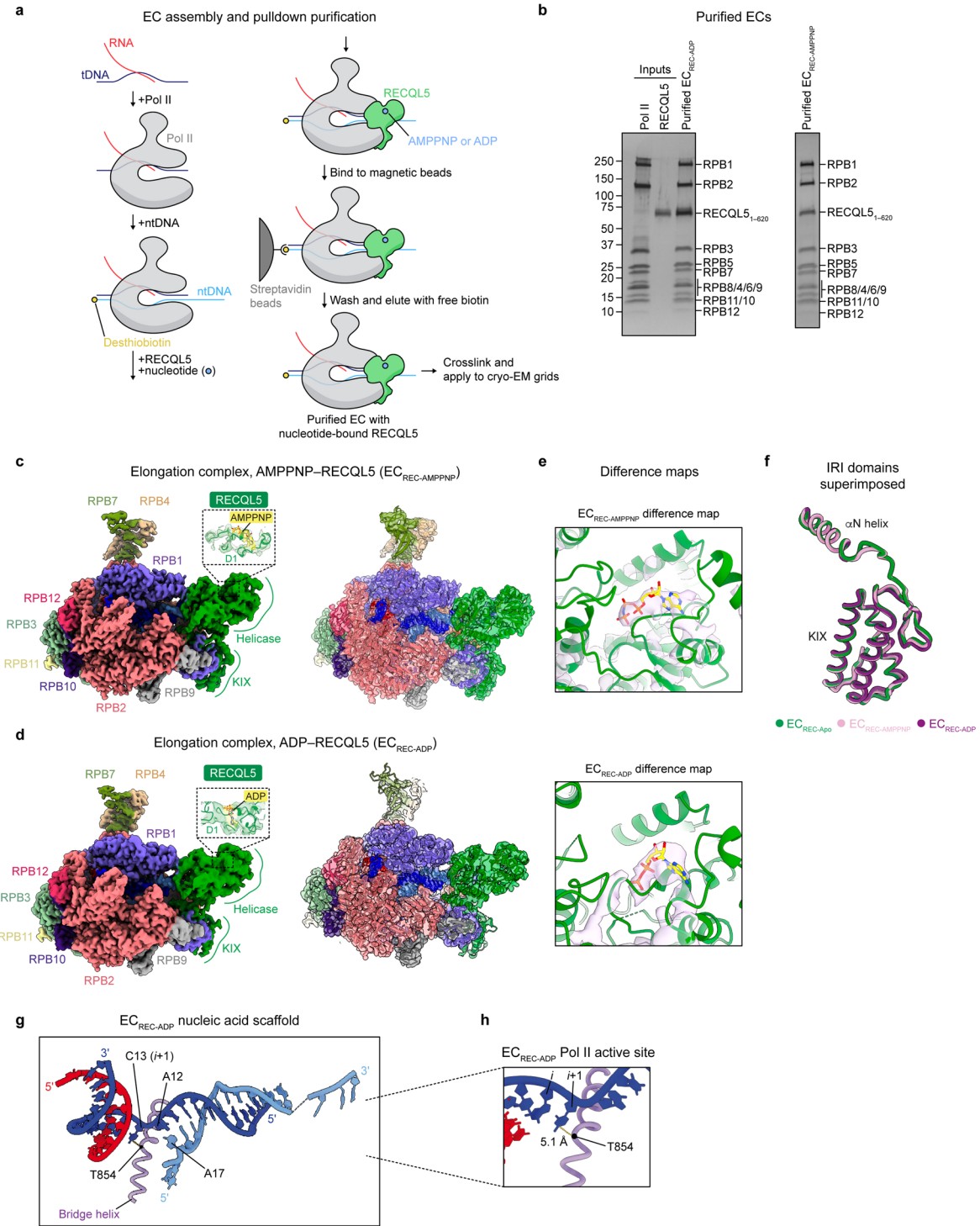

**Extended Data Fig. 7 | Structures of ECs with nucleotide-bound RECQL5 helicase. a**, Schematic depicting pulldown strategy to reconstitute and purify Pol II ECs with nucleotide-bound RECQL5. **b**, SDS-PAGE analysis of pulldown inputs and the final purified $EC_{REC-ADP}$ (left) and $EC_{REC-AMPPNP}$ (right). Proteins were visualized by silver staining. SDS-PAGE analysis was performed once. **c**, Cryo-EM map (left) and cross-section with fitted atomic model (right) of Pol II EC bound to $RECQL5_{1-620}$ and AMPPNP ($EC_{REC-AMPPNP}$). Pol II subunits are colored as indicated by the labels, and nucleic acids are colored as in Fig. 1b. The inset shows the map and model around the helicase nucleotide binding site. **d**, Cryo-EM map (left) and cross-section with fitted atomic model (right) of Pol II EC bound to $RECQL5_{1-620}$

and ADP ($EC_{REC-ADP}$). Colors are the same as in **c**. The inset shows the map and model around the helicase nucleotide binding site. **e**, Difference maps for $EC_{REC-AMPPNP}$ (top) and $EC_{REC-ADP}$ (bottom). View shows the RECQL5 helicase nucleotide binding site. The helicase is depicted in green, and the modeled nucleotide in yellow. Details in Methods. **f**, Superposition of the RECQL5 IRI modules (αN helix and KIX domain) for the $EC_{REC-Apo}$ (green), $EC_{REC-AMPPNP}$ (pink), and $EC_{REC-ADP}$ (purple) structures. **g**, Model of the nucleic acid scaffold and Pol II bridge helix in the $EC_{REC-ADP}$ structure. Important nucleotides are highlighted. **h**, Close up of the $EC_{REC-ADP}$ Pol II active site shown in **e** highlighting the distance between the $i+1$ site nucleotide base and RPB1 T854 in the bridge helix.

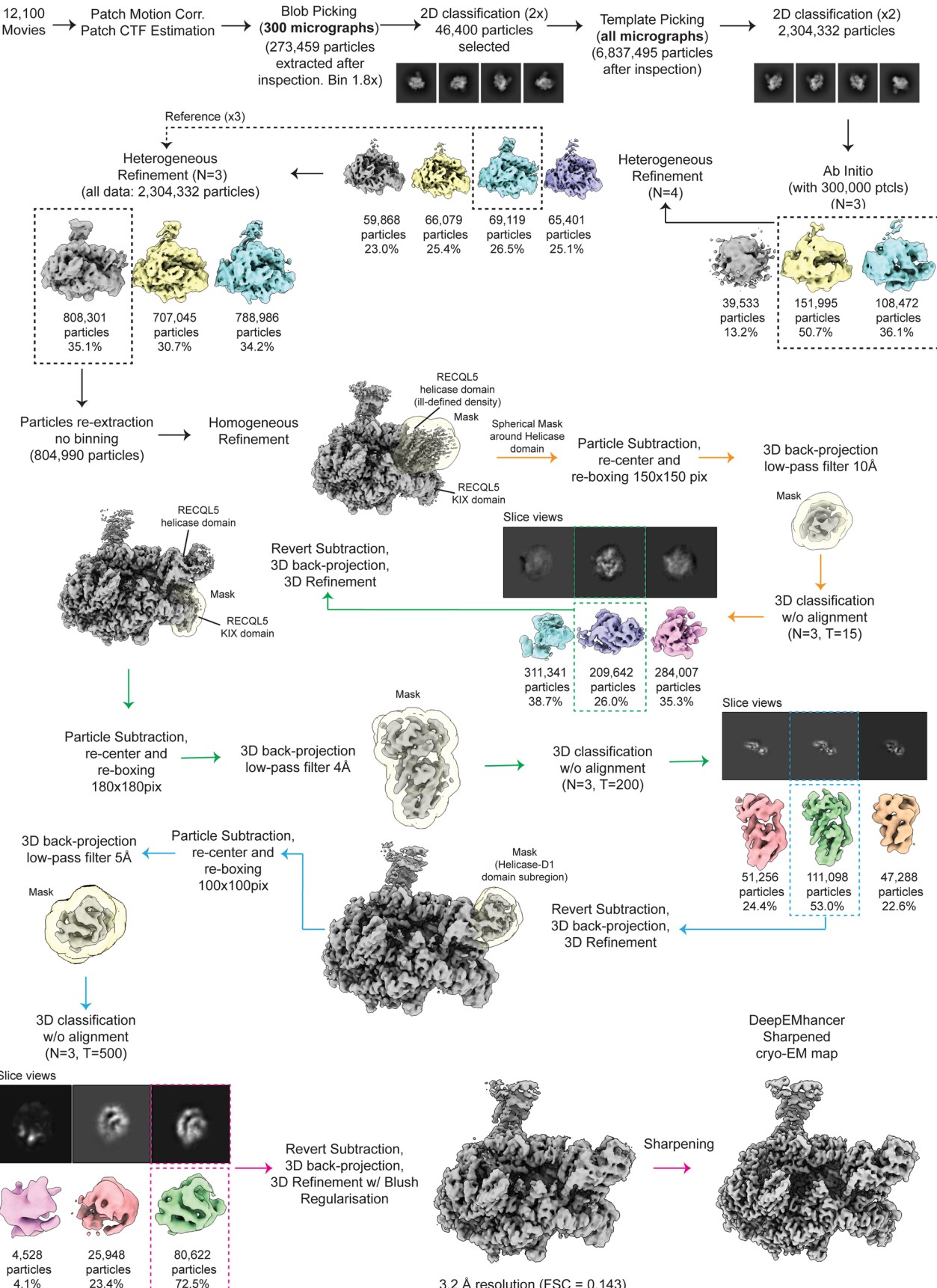

**Extended Data Fig. 8 | Cryo-EM processing workflow for EC_REC-AMPPNP.** Workflow showing cryo-EM processing steps for the EC_REC-AMPPNP structure.

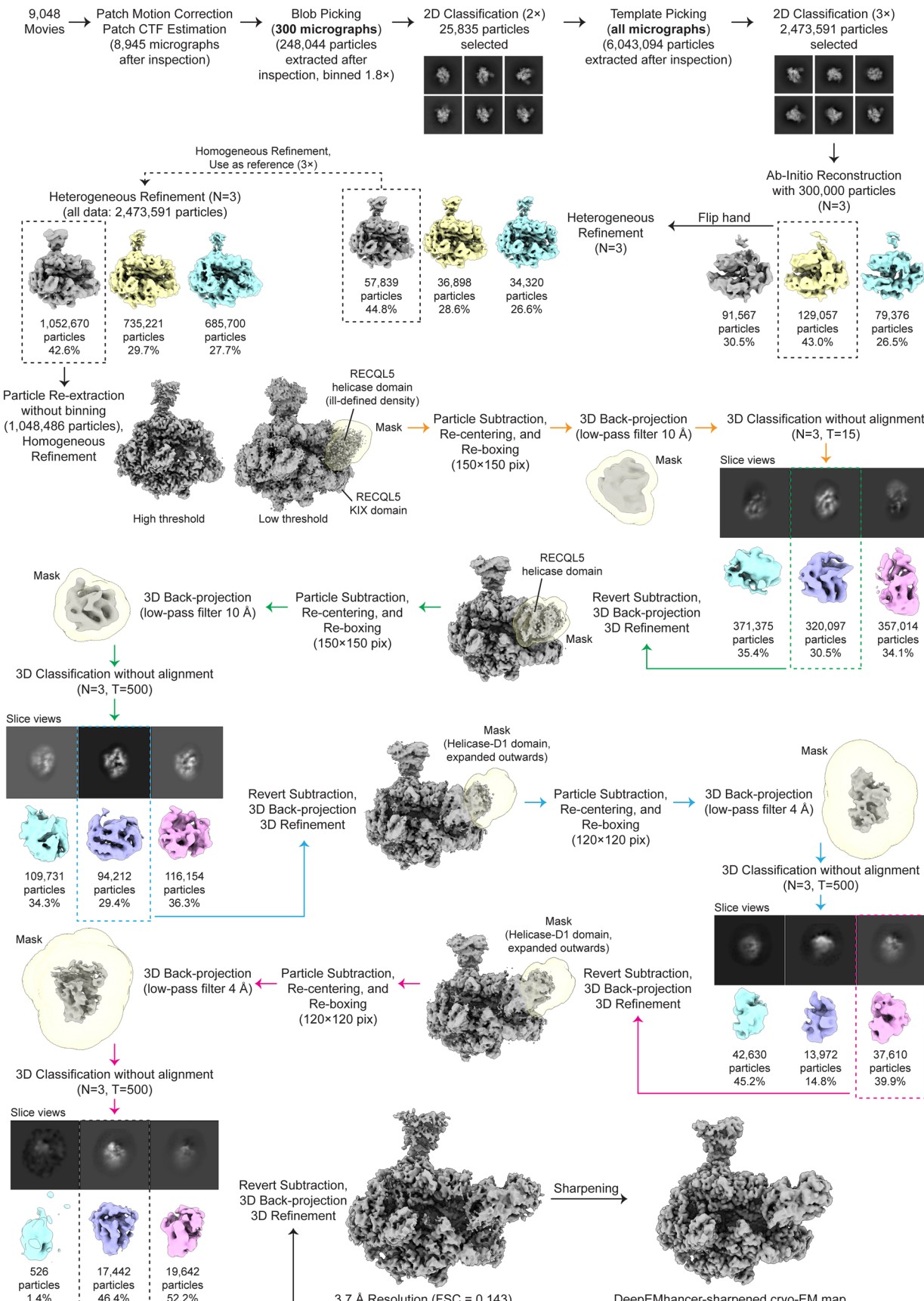

**Extended Data Fig. 9 | Cryo-EM processing workflow for EC<sub>REC-ADP</sub>.** Workflow showing cryo-EM processing steps for the EC$_{REC-ADP}$ structure.

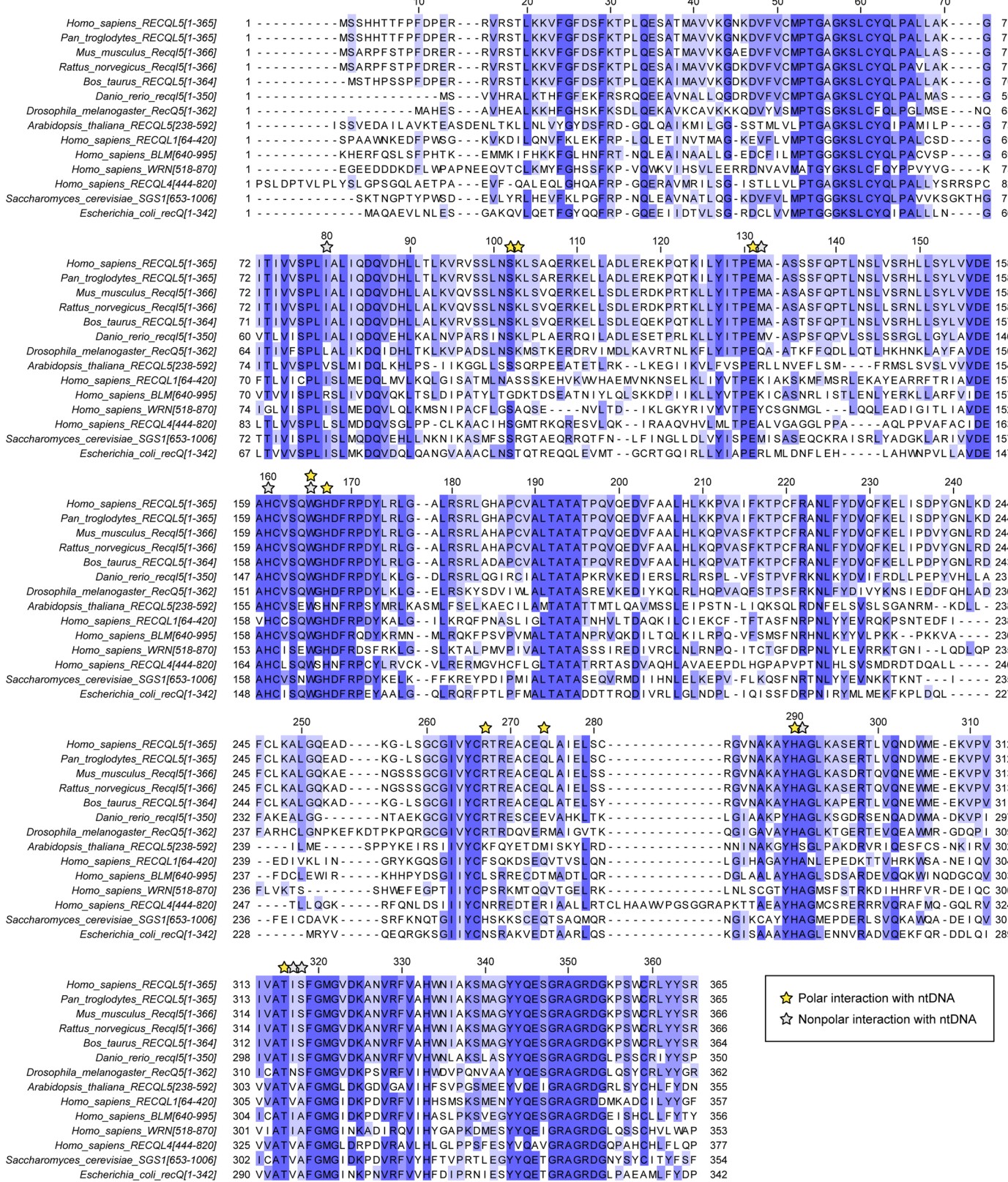

**Extended Data Fig. 10 | Conservation of helicase domain residues across RECQL5 orthologs and other RecQ family members.** Sequence alignment and conservation of helicase domains using human RECQL5 residues 1–365 as a reference. Proteins included in this analysis include RECQL5 orthologs (from *H. sapiens*, *P. troglodytes*, *M. musculus*, *R. norvegicus*, *B. taurus*, *D. rerio*, *D.*

*melanogaster*, and *A. thaliana*) as well as other RecQ family members (RECQL1, BLM, WRN, and RECQL4 from *H. sapiens*, SGS1 from *S. cerevisiae*, and recQ from *E. coli*). Numbers above the sequences mark every ten residues in the human RECQL5 sequence. Alignment was conducted using Clustal Omega through EMBL-EBI[54] and visualized using Jalview[55].

# Reporting Summary

## Statistics

For all statistical analyses, confirm that the following items are present in the figure legend, table legend, main text, or Methods section.

| n/a | Confirmed | |
|---|---|---|
| ☐ | ☒ | The exact sample size (*n*) for each experimental group/condition, given as a discrete number and unit of measurement |
| ☒ | ☐ | A statement on whether measurements were taken from distinct samples or whether the same sample was measured repeatedly |
| ☒ | ☐ | The statistical test(s) used AND whether they are one- or two-sided<br>*Only common tests should be described solely by name; describe more complex techniques in the Methods section.* |
| ☒ | ☐ | A description of all covariates tested |
| ☒ | ☐ | A description of any assumptions or corrections, such as tests of normality and adjustment for multiple comparisons |
| ☒ | ☐ | A full description of the statistical parameters including central tendency (e.g. means) or other basic estimates (e.g. regression coefficient) AND variation (e.g. standard deviation) or associated estimates of uncertainty (e.g. confidence intervals) |
| ☒ | ☐ | For null hypothesis testing, the test statistic (e.g. *F*, *t*, *r*) with confidence intervals, effect sizes, degrees of freedom and *P* value noted<br>*Give P values as exact values whenever suitable.* |
| ☒ | ☐ | For Bayesian analysis, information on the choice of priors and Markov chain Monte Carlo settings |
| ☒ | ☐ | For hierarchical and complex designs, identification of the appropriate level for tests and full reporting of outcomes |
| ☒ | ☐ | Estimates of effect sizes (e.g. Cohen's *d*, Pearson's *r*), indicating how they were calculated |

*Our web collection on statistics for biologists contains articles on many of the points above.*

## Software and code

Policy information about availability of computer code

| | |
|---|---|
| Data collection | SerialEM (v4.1.0beta) was used for cryo-EM data collection. Typhoon FLA 9500 (v1.1) was used to scan gels for the RNA extension assay. |
| Data analysis | cryoSPARC (v4.5.3) and RELION (v5) were used for cryo-EM data processing. Coot (v0.9.8.7) and Phenix (v1.20) were used for model building and refinement. AlphaFold (v3) was used to generate an initial model for part of RECQL5. ChimeraX (v1.8) was used to visualize and interpret structures. Python (v3.9.6) was used to visualize FSC curve plots from RELION and Phenix output files. Adobe Photoshop 2023 was used to process RNA extension assay raw images (only adjustments to tonal range were performed, and all adjustments were applied to the whole image). Clustal Omega (v1.2.4) (through EMBL-EBI, https://www.ebi.ac.uk/jdispatcher/msa/clustalo) and Jalview (v2.11.4.1) were used to align sequences and visualize evolutionary conservation, respectively. Adobe Illustrator 2023 was used to create final figures. |

For manuscripts utilizing custom algorithms or software that are central to the research but not yet described in published literature, software must be made available to editors and reviewers. We strongly encourage code deposition in a community repository (e.g. GitHub). See the Nature Portfolio guidelines for submitting code & software for further information.

## Data

Policy information about availability of data

All manuscripts must include a data availability statement. This statement should provide the following information, where applicable:
- Accession codes, unique identifiers, or web links for publicly available datasets
- A description of any restrictions on data availability
- For clinical datasets or third party data, please ensure that the statement adheres to our policy

All data pertaining to this paper are provided within the paper or accessible from public repositories. The cryo-EM density maps and their respective atomic coordinate files have been deposited to the Electron Microscopy Data Bank (EMDB) and Protein Data Bank (PDB) under the following accession codes: EMD-48071 and PDB 9EHZ (ECFree), EMD-48073 and PDB 9EI1 (ECREC-Apo), EMD- 48074 and PDB 9EI2 (ECREC-Apo (IRI Focused)), EMD- 48075 and PDB 9EI3 (ECREC-AMPPNP), and EMD-48076 and PDB 9EI4 (ECREC-ADP). Raw cryo-EM movies have been deposited to the Electron Microscopy Public Image Archive (EMPIAR) under the following accession codes: EMPIAR-12711 (dataset for ECFree, ECREC-Apo, and ECREC-Apo (IRI Focused)), EMPIAR-12721 (dataset for ECREC-AMPPNP), and EMPIAR-12722 (dataset for ECREC-ADP). In addition to the structures reported in this work, the following publicly available structures were used: PDB accession codes 2VUM, 1I6H, 6EXV, 5FLM, 5LB8, 8JH2, and 7NW0. Source data are provided with the manuscript.

## Research involving human participants, their data, or biological material

Policy information about studies with human participants or human data. See also policy information about sex, gender (identity/presentation), and sexual orientation and race, ethnicity and racism.

| | |
|---|---|
| Reporting on sex and gender | N/A |
| Reporting on race, ethnicity, or other socially relevant groupings | N/A |
| Population characteristics | N/A |
| Recruitment | N/A |
| Ethics oversight | N/A |

Note that full information on the approval of the study protocol must also be provided in the manuscript.

# Field-specific reporting

Please select the one below that is the best fit for your research. If you are not sure, read the appropriate sections before making your selection.

☒ Life sciences  ☐ Behavioural & social sciences  ☐ Ecological, evolutionary & environmental sciences

For a reference copy of the document with all sections, see nature.com/documents/nr-reporting-summary-flat.pdf

# Life sciences study design

All studies must disclose on these points even when the disclosure is negative.

| | |
|---|---|
| Sample size | Cryo-EM datasets were collected such that the number of particles in final classes were sufficient to yield high-resolution structures, based on empirical findings and prior experience. For the RNA extension assay, we performed this in singlicate but replicated the results in 3 independent experiments, which is standard in the field (for example, see Su & Vos, Mol. Cell 84, 1243, 2024). Other miscellaneous gels were run once, which is standard in the field and acceptable as they are only for sample validation purposes. |
| Data exclusions | Micrographs and particles were discarded following criteria that are standard to the field (e.g., micrographs with poor CTF fit resolution and particles corresponding to graphene oxide edges and water ice were discarded). We also performed detailed 3D classification to select particles belonging to states of interest. 3D classification is standard in the field. All details are provided in the Methods and data processing figures (Extended Data Figs. 1, 3, 5, 8, and 9). |
| Replication | Cryo-EM structures were not replicated, which is standard. RNA extension assay was performed 3 times with consistent results, which is standard in the field. |
| Randomization | Not applicable to our study. None of the experiments conducted involved dividing samples/organisms/participants into experimental groups. For purposes of cryo-EM validation, particles were randomly divided into two half-sets during image processing following established standards in the field ("gold-standard" FSC procedure). Randomization was performed using cryoSPARC, a standard program used in the field. |
| Blinding | Investigators were not blinded, which is standard for the cryo-EM field. Blinding is not routine in structural/biochemical studies, in part due to its impracticality in the context of structural/biochemical procedures and also since in many cases it is important to know sample identity when preparing cryo-EM samples, collecting cryo-EM data, and processing cryo-EM data. |

# Reporting for specific materials, systems and methods

We require information from authors about some types of materials, experimental systems and methods used in many studies. Here, indicate whether each material, system or method listed is relevant to your study. If you are not sure if a list item applies to your research, read the appropriate section before selecting a response.

## Materials & experimental systems

| n/a | Involved in the study |
|---|---|
| ☐ | ☒ Antibodies |
| ☐ | ☒ Eukaryotic cell lines |
| ☒ | ☐ Palaeontology and archaeology |
| ☒ | ☐ Animals and other organisms |
| ☒ | ☐ Clinical data |
| ☒ | ☐ Dual use research of concern |
| ☒ | ☐ Plants |

## Methods

| n/a | Involved in the study |
|---|---|
| ☒ | ☐ ChIP-seq |
| ☒ | ☐ Flow cytometry |
| ☒ | ☐ MRI-based neuroimaging |

## Antibodies

| | |
|---|---|
| Antibodies used | Anti-RPB1 antibody (BioLegend, cat. no. 920102, Clone 8WG16, Lot B217159, 500 uL antibody for 1 prep of Pol II from 114 L HeLa cells) was used for purification of endogenous Pol II. |
| Validation | Antibody specificity and performance was validated in this study by testing the purity of the purified Pol II by SDS-PAGE, verifying purified Pol II by negative-stain electron microscopy, and ultimately visualizing high-resolution structures by cryo-EM. The final structures confirm that the purified Pol II contains all subunits. |

## Eukaryotic cell lines

Policy information about cell lines and Sex and Gender in Research

| | |
|---|---|
| Cell line source(s) | HeLa cells were obtained from the UC Berkeley Cell Culture Facility, and were used only for purification of endogenous human Pol II and not for any functional experiments. |
| Authentication | HeLa cells were authenticated by the UC Berkeley Cell Culture Facility using a GenePrint 10 kit (Promega) for short tandem repeat processing. |
| Mycoplasma contamination | HeLa cells were tested for mycoplasma by the UC Berkeley Cell Culture Facility and were negative. |
| Commonly misidentified lines (See ICLAC register) | No commonly misidentified cell lines were used in this study. |

## Plants

| | |
|---|---|
| Seed stocks | N/A |
| Novel plant genotypes | N/A |
| Authentication | N/A |

