## [Peer Review File · Nature Structural & Molecular Biology]

Structural insights into transcriptional regulation by the helicase RECQL5

Corresponding Author: Professor Eva Nogales

Version 0:

Decision Letter:

21st Jan 2025

Dear Professor Nogales,

Thank you again for submitting your manuscript "Structural insights into transcriptional regulation by the helicase RECQL5". We now have comments (below) from the 2 reviewers who evaluated your paper. In light of those reports, we remain interested in your study and would like to see your response to the comments of the referees, in the form of a revised manuscript.

You will see that while both reviewers appreciate the work, Reviewer #1 requests discussion of how the use of a non-standard nucleic acid may contribute to the observed structural states, and to comment on when the RECQL5-stabilizing substrate with single-stranded downstream DNA would be present in cells, echoed by Reviewer #2. Moreover, Reviewer #1 requests additional functional data to support the structural work, including RNA extension assays from the employed scaffold, and nucleotide incorporation in RNA when RECQL5 is given AMP-PNP as a substrate. Editorially, we agree that these, and potentially other, functional corroborations would notably strengthen the manuscript, and would therefore require their addition before we consider a revised manuscript. Please be sure to address/respond to all concerns of the referees in full in a point-by-point response and highlight all changes in the revised manuscript text file.

We appreciate the requested revisions are extensive. We thus expect to see your revised manuscript within 3 months. If you cannot send it within this time, please let us know. We will be happy to consider your revision as long as nothing similar has been accepted for publication at NSMB or published elsewhere. Should your manuscript be substantially delayed without notifying us in advance and your article is eventually published, the received date would be that of the revised, not the original, version.

Reporting Summary:

EXTENDED DATA FIGURES

When re-submitting your manuscript, please ensure that any supplementary figures and tables that are crucial to the

manuscript's conclusions are converted into Extended Data figures and tables to increase visibility of these data. Extended Data figures and tables are online-only (present in the online PDF and full-text HTML versions of the paper), peer-reviewed display items that provide essential background to the article but are not included in the main article due to space constraints. A maximum of ten Extended Data display items (figures and tables) is permitted.

Please note that all key data shown in the main figures as cropped gels or blots should be presented in uncropped form, with molecular weight markers. These data can be aggregated into a single supplementary figure. While these data can be displayed in a relatively informal style, they must refer back to the relevant figures. These data should be submitted with the last revision, prior to acceptance, but you may want to start putting it together at this point.

SOURCE DATA: we request that authors provide, in tabular form, the data underlying the graphical representations used in figures. This is to further increase transparency in data reporting, as detailed in this editorial (<http://www.nature.com/nsmb/journal/v22/n10/full/nsmb.3110.html>). Spreadsheets can be submitted in excel format. Only one (1) file per figure is permitted; thus, for multi-paneled figures, the source data for each panel should be clearly labeled in the Excel file; alternately the data can be provided as multiple, clearly labeled sheets in an Excel file. When submitting files, the title field should indicate which figure the source data pertains to. We encourage our authors to provide source data at the revision stage, so that they are part of the peer-review process.

We require deposition of coordinates (and, in the case of crystal structures, structure factors) into the Protein Data Bank with the designation of immediate release upon publication (HPUB). Electron microscopy-derived density maps and coordinate data must be deposited in EMDB and released upon publication. Deposition and immediate release of NMR chemical shift assignments are highly encouraged. Deposition of deep sequencing and microarray data is mandatory, and the datasets must be released prior to or upon publication. To avoid delays in publication, dataset accession numbers must be supplied with the final accepted manuscript and appropriate release dates must be indicated at the galley proof stage. Please find the complete NRG policies on data availability at <http://www.nature.com/authors/policies/availability.html>.

Link Redacted

Sincerely,
Sara

Sara Osman, Ph.D.
Senior Editor
Nature Structural & Molecular Biology

Referee expertise:

Referee #1: Structural biology, transcription

Referee #2: Structural biology, helicases

Reviewers' Comments:

Reviewer #1 (Remarks to the Author):

Here Ariza, Lue et al., determine high resolution cryo-EM structures of RNA polymerase II elongation complexes in isolation and in complex with the helicase, RECQL5. Previous structures of RECQL5 were of insufficient resolution to identify specific interactions between RNA polymerase II and RECQL5. Using modern data collection strategies, the authors have overcome these limitations. The authors reconstitute RECQL5 in both nucleotide complexed and apo conformations, and the authors capture an unusual translocation state in the absence of nucleotide or RECQL5. In the presence of AMP-PNP, the polymerase active site is able to adopt the canonical post-translocation state, and it appears that this is supported by RECQL5 function. The manuscript is well written and the structural data is of high quality. I have a few comments the authors should address prior to publication.

Major points

1-The authors use a non-standard nucleic scaffold to form a transcription elongation complex suitable for cryo-EM studies (comparing to PDBs such as 5OIK, 6exv). In particular, the RNA-DNA hybrid is longer than in most elongation complexes (typically 8-10 base pair RNA-DNA hybrid, here it is 11 bp), the upstream DNA is quite short (3 nts, typically 10-14 nts), and the non-template DNA bubble is on the larger side (13 nts). The authors should comment on how this may contribute to the interesting translocation intermediate they observe. The longer RNA-DNA hybrid could be classified as an over-extended hybrid, which may interfere with transcription elongation. The reviewer notes that this appears to be the same scaffold that was used in Kassube et al., 2013.

2-The authors use a single stranded DNA on the non-template, downstream DNA to stabilize the RECQL5 helicase, as they did in their prior study. It would be helpful if the authors could comment on when they would expect this type of substrate to be present in cells. They mention that RECQL5 is important for evicting RAD51 from ssDNA, however, it is unclear if this function would be related to its role in transcription. In their previous work, they hypothesize this could be a part of MRN function. Given the time between these studies, it would help to restate their model.

3-The authors should provide some functional data to support their beautiful structural work. For the above point 1, the template may be challenging for Pol II to transcribe, and RNA extension assays on this exact scaffold would show if this is the case. Similarly, it would be expected from the provided structural work that RNA polymerase II would be able to incorporate 1 nt into the RNA when RECQL5 is given AMP-PNP as a substrate. Finally, given the high resolution of the data and reported interactions, it would help if the authors could test the importance of the additional interactions RECQL5 makes with the NT DNA in context of RNA extension assays. This supporting biochemical work would strengthen the authors conclusions.

Minor comment

1- All structural models appear to have Ramachandran outliers. It would be best to remove/improve these outliers prior to publication, given the high quality of the EM data.

2-The authors may want to state that most elongation factors with the exception of TFIIIS would likely be able to associate with a RECQL5 bound Pol II, and initiation factors like TFIIH would likely not be able to form a complex with RECQL5 bound Pol II. This would be helpful for those studying RECQL5 function at different stages of transcription.

Reviewer #2 (Remarks to the Author):

The manuscript from the Nogales group is a tour de force, providing abundant, high-quality structural data which examine in detail the functional interactions between an elongating RNA polymerase II and the helicase RECQL5. RECQL5 has been shown previously to slow down transcription by RNAPII in vitro, and to be important in preventing DNA damage following replicative or transcriptional stress. The paper presents single-particle EM structures of an RNAPII elongation complex, alone and in combination with truncated RECQL5 in nucleotide-free, AMP-PNP and ADP forms. These structures show the binding determinants between RNAPII, RECQL5, and the nucleic acids. Combined with previous structural analyses of RNAPII conformations, these structures paint a vivid picture of the mechanism by which RECQL5 slows transcription and helps to bypass transcriptional blocks, while alternating between apo-, ATP and ADP-bound states.

The results are technically remarkable. The authors cleverly classify and refine individual molecular images from heterologous grids to achieve high resolution within the polymerase active site, allowing to precisely define the register of the polymerase on the RNA/DNA structure. The resolution of the helicase domains is sufficient to observe the position of bound ATP and the domain arrangement; Compared to the apo- and ATP structures, the observation of a less ordered state of the helicase domains when bound to ADP gains significance.

The conclusions from the structural analysis are convincing and the authors are careful not to over-interpret the results. The description and the figures are clear and well-explained-

I cannot really find any faults with this manuscript, and it should be published rapidly.

One small question I have: the DNA/RNA substrate used in the structural analysis includes a 3' single-stranded DNA

extension on the non-transcribed strand, which is the docking site for binding of RECQL5; the binding mode seems to prefer ssDNA. Could the authors comment on what this ssDNA region may represent in a genomic environment, and whether there is any biochemical data to support a role of a ssDNA tail for the action of RECQL5.

Version 1:

Decision Letter:

Our ref: NSMB-A50225A

17th Apr 2025

Dear Dr. Nogales,

Thank you for submitting your revised manuscript "Structural insights into transcriptional regulation by the helicase RECQL5" (NSMB-A50225A). It has now been seen by the original referees and their comments are below. The reviewers find that the paper has improved in revision, and therefore we'll be happy in principle to publish it in Nature Structural & Molecular Biology, pending minor revisions to satisfy the referees' final requests and to comply with our editorial and formatting guidelines.

We are now performing detailed checks on your paper and will send you a checklist detailing our editorial and formatting requirements within the next few weeks. Please do not upload the final materials and make any revisions until you receive this additional information from us.

To facilitate our work at this stage, it is important that we have a copy of the main text as a word file. If you could please send along a word version of this file as soon as possible, we would greatly appreciate it; please make sure to copy the NSMB account (cc'ed above).

Sincerely,
Sara

Sara Osman, Ph.D.
Senior Editor
Nature Structural & Molecular Biology

Reviewer #1 (Remarks to the Author):

The authors have addressed all of my concerns.

Reviewer #2 (Remarks to the Author):

The revised manuscript has addressed the questions I raised and presents a much clearer view of the interactions of the polymerase and associated factors with the DNA/RNA. I recommend publication of the paper.

Reviewer #1:

Here Ariza, Lue et al., determine high resolution cryo-EM structures of RNA polymerase II elongation complexes in isolation and in complex with the helicase, RECQL5. Previous structures of RECQL5 were of insufficient resolution to identify specific interactions between RNA polymerase II and RECQL5. Using modern data collection strategies, the authors have overcome these limitations. The authors reconstitute RECQL5 in both nucleotide complexed and apo conformations, and the authors capture an unusual translocation state in the absence of nucleotide or RECQL5. In the presence of AMP-PNP, the polymerase active site is able to adopt the canonical post-translocation state, and it appears that this is supported by RECQL5 function. The manuscript is well written and the structural data is of high quality. I have a few comments the authors should address prior to publication.

We thank the reviewer for their constructive feedback.

Major points

1-The authors use a non-standard nucleic acid scaffold to form a transcription elongation complex suitable for cryo-EM studies (comparing to PDBs such as 5OIK, 6exv). In particular, the RNA-DNA hybrid is longer than in most elongation complexes (typically 8-10 base pair RNA-DNA hybrid, here it is 11 bp), the upstream DNA is quite short (3 nts, typically 10-14 nts), and the non-template DNA bubble is on the larger side (13 nts). The authors should comment on how this may contribute to the interesting translocation intermediate they observe. The longer RNA-DNA hybrid could be classified as an over-extended hybrid, which may interfere with transcription elongation. The reviewer notes that this appears to be the same scaffold that was used in Kassube et al., 2013.

The nucleic acid scaffold we used is the same as was used in our previous study¹, which itself is a truncated version of the scaffold used in an earlier crystallographic study of the Pol II elongation complex² from the Cramer lab. As designed, our scaffold should have only 8 bp of RNA-DNA hybrid, with a 6 bp upstream DNA hybrid, as shown in **Fig. A (top)**. However, in our EC_{REC-Apo} structure, we observed an additional three bases of RNA positioned close to their corresponding tDNA bases. This appears to extend the RNA-DNA hybrid up to 11 bp despite the resulting A-A mismatch, as shown in **Fig. A (bottom)**. It is possible that our shortening of the upstream DNA region (compared to in ref. ²) could have led to disfavoring of upstream DNA pairing in favor of an extended RNA-DNA hybrid. We opted to depict the scaffold showing the 11 bp RNA-DNA hybrid in our schematic in **Fig. 1b** in order to reflect the empirically observed structure.

Fig. A. Different representations of our nucleic acid scaffold.

The reviewer brings up another good point, which is that the scaffold we used could account for the fact that we observe Pol II to adopt an intermediate translocation state in EC_{Free}. Therefore, we have added text commenting on this matter (lines 238–240). Importantly, all of our structural work was performed using the same nucleic acid scaffold, justifying our comparison of Pol II translocation state across structures.

With respect to the reviewer's concern that the over-extended hybrid could interfere with transcription, we now report RNA extension assay data confirming

that transcription on this scaffold is possible (see below).

2-The authors use a single stranded DNA on the non-template, downstream DNA to stabilize the RECQL5 helicase, as they did in their prior study. It would be helpful if the authors could comment on when they would expect this type of substrate to be present in cells. They mention that RECQL5 is important for evicting RAD51 from ssDNA, however, it is unclear if this function would be related to its role in transcription. In their previous work, they hypothesize this could be a part of MRN function. Given the time between these studies, it would help to restate their model.

The reviewer brings up an important point regarding the single-stranded DNA used in our study and its relevance in cells. There are several situations where Pol II could encounter a downstream single-stranded non-template DNA (ntDNA) bound by RECQL5. R-loops are known to form as a result of transcription and consist of a template DNA (tDNA)-RNA hybrid with a displaced single-stranded ntDNA³. An architecture similar to the one we investigated structurally would occur if a transcribing Pol II advanced toward and collided with an R-loop formed by a second Pol II transcribing the same gene ahead of it. Alternatively, RECQL5 could bind to the single-stranded ntDNA at an R-loop and, through its 3' to 5' helicase activity, move upstream until it collided with a transcribing Pol II in a head-to-head collision. As the reviewer points out, RECQL5 is also known to interact with the MRN complex, a double-stranded break (DSB) sensor, which helps to recruit RECQL5 to DSB sites⁴. Loading of RECQL5 at these sites and subsequent 3' to 5' translocation away from the DSB site could also result in a situation where Pol II encounters a single-stranded ntDNA region downstream of it bound to RECQL5. Finally, in head-on transcription-replication conflicts, the ntDNA downstream of Pol II is the leading strand of the replication fork. We speculate that, given RECQL5's known role in clearing RAD51 filaments from these stalled replication forks⁵, it could be possible for RECQL5 to bind to the single-stranded region of the leading strand and translocate upstream to contact Pol II.

To address the reviewer's request, we have modified our Discussion section to include discussion of these points (lines 508–521). We have also placed our new findings in this study in the context of our previous hypothesis that RECQL5 could aid in protecting genome stability at DSB sites in tandem with the MRN complex (lines 468–470). Additionally, with respect to transcription-replication conflicts and RECQL5's role in evicting RAD51, we mention that it is an important outstanding question to determine if it is related to its role in transcriptional regulation (lines 521–525).

3-The authors should provide some functional data to support their beautiful structural work. For the above point 1, the template may be challenging for Pol II to transcribe, and RNA extension assays on this exact scaffold would show if this is the case. Similarly, it would be expected from the provided structural work that RNA polymerase II would be able to incorporate 1 nt into the RNA when RECQL5 is given AMP-PNP as a substrate. Finally, given the high resolution of the data and reported interactions, it would help if the authors could test the importance of the additional interactions RECQL5 makes with the NT DNA in context of RNA extension assays. This supporting biochemical work would strengthen the authors conclusions.

We thank the reviewer for their comments. We have now conducted RNA extension assays to interrogate the effect of RECQL5 on Pol II elongation. In the revised manuscript, we present new functional data in **Fig. 4h**. Below, we provide a summary of the new experiments and our interpretation of the results.

To address the reviewer's first point, we tested Pol II activity on the nucleic acid scaffold employed for structural studies (**Fig. 4h**). These results confirm that Pol II can transcribe on this scaffold efficiently. Starting from a 20 nt initial RNA, the strongest two top bands appear to correspond to 28–29 nt transcripts. Furthermore, addition of RECQL5 to these reactions leads to inhibition of RNA extension as expected, as shown by (1) the formation of smaller products (22–25 nt) not observed in the absence of RECQL5,

consistent with RECQL5 serving as a roadblock, as well as (2) a large decrease in the production of longer (28–29 nt) RNA transcripts.

Fig. B. Single-nucleotide RNA extension assay testing the effect of AMPPNP on Pol II incorporation of a single G in the presence of RECQL5.

expectations, no effect was observed when RECQL5 was supplied with AMPPNP (**Fig. B**). However, there are a few caveats to this experiment. From a technical perspective, the rate of single-nucleotide RNA extension from the initial pre-assembled ECs (as opposed to later in the reaction after Pol II dissociates and binds to a new substrate) may still have been too fast for us to detect in our bulk assays. Also, it is possible that the intermediate translocation state we observed for EC_{REC-Apo} could still accommodate an incoming GTP, and the difference in rates is something we are unable to capture. Finally, in order to slow the reaction down enough to capture, we had to greatly reduce the concentration of GTP to far below physiological concentrations, and only 5-fold higher than the RNA substrate itself. Thus, we believe it is difficult to rule out an effect of AMPPNP on RECQL5-dependent transcriptional inhibition based on these results. We note that rigorous testing would require specialized equipment (i.e., rapid kinetics quench-flow experiments) that are beyond the scope of our current paper, but could be a subject for future studies.

However, our original assay (evaluating transcription on the full-length template) does provide some evidence in support of a positive effect of RECQL5 on RNA extension. As we describe in the text, we also observed a band (at 30 nt) larger than the major 28–29 nt products both when RECQL5 is added and omitted (**Fig. 4h**). Interestingly, RECQL5 appeared to promote the conversion of the 28–29 nt products to this 30 nt product. This is shown by the disappearance of the 29 nt band in the presence of RECQL5 and the much higher ratio between the 30 nt product to its 28–29 nt precursors in the presence of RECQL5. In these assays, 0.8 mM ATP is provided for RNA extension, and therefore RECQL5 should be largely nucleotide-bound. Thus, these results suggest that RECQL5, in its nucleotide-bound state, can promote an additional RNA extension, which motivate our proposal that RECQL5's helicase domain, through modulating Pol II's translocation state, can reduce transcription stress.

We have modified our Discussion section in our revised manuscript (including the text in **Fig. 5**) to soften our language with regards to our proposed model and frame it as a hypothesis that should be explored by further studies. Additionally, we include new text suggesting how future studies could interrogate RECQL5's effect on Pol II translocation and its biological significance, for example using single-molecule optical tweezer approaches (lines 501–506).

Finally, we appreciate the reviewer's suggestion to test how RECQL5–ntDNA interactions contribute to RECQL5's effects on RNA extension. Given the dynamic nature of RECQL5's helicase domain, it is possible that mutating certain DNA-interacting residues could lead to no effect on transcription inhibition, or an effect that is difficult to interpret. Purifying such mutants, in addition to extra Pol II, and testing these in RNA extension assays would require a significant investment of resources. While such experiments would be interesting, we believe that they are not critical to our paper, which focuses on the overall architecture of the RECQL5-bound EC and the effects of RECQL5 on the Pol II translocation state. Given the strength of our structural results in addition to the new biochemical data, we believe that these experiments could be left for

future studies. However, to address the reviewer's request to assess the relevance of the RECQL5-ntDNA contacts presented in our study, we used bioinformatic tools and conducted an analysis of evolutionary conservation of the helicase domain, which is presented in a new figure, **Extended Data Fig. 8**, and discussed in lines 378–386. This analysis revealed that many of the RECQL5 helicase residues we identified to mediate salt bridges or polar interactions with the non-template single-stranded DNA are in fact highly conserved, both among RECQL5 orthologs from a diverse array of organisms and among other RecQ family members in humans, *S. cerevisiae*, and *E. coli*. Interestingly, two residues, K103 and R267, appear to be highly conserved among RECQL5 orthologs in animals, but are not well conserved in other RecQ family members. This could point toward an evolutionarily conserved, RECQL5-specific role such as transcription inhibition. However, it should be noted that, to our knowledge, transcriptional inhibition has only been experimentally verified for human RECQL5 so far. Overall, our conservation analysis provides evidence that these ntDNA-interacting residues are important for RECQL5's function.

Minor comment

1- All structural models appear to have Ramachandran outliers. It would be best to remove/improve these outliers prior to publication, given the high quality of the EM data.

We appreciate the reviewer's detailed inspection of our structural data and their assessment that it is of high quality. With respect to the Ramachandran outliers, these are all located in peripheral regions of Pol II or in low-resolution areas of the RECQL5 helicase. For these residues, the corresponding density is not well defined, and side chains are not observed. Therefore, we felt that additional refinement to lower the number of outliers may not be appropriate, as it would lead to over-refinement of the structures that are already of good quality. We would like to mention that the number of outliers account for no more than 25 out of ~4,000 residues in the Pol II–RECQL5 complexes (0.2%–0.7% total residues), and do not correspond to any molecular contacts or notable features discussed in this manuscript.

2-The authors may want to state that most elongation factors with the exception of TFIIS would likely be able to associate with a RECQL5 bound Pol II, and initiation factors like TFIIH would likely not be able to form a complex with RECQL5 bound Pol II. This would be helpful for those studying RECQL5 function at different stages of transcription.

We appreciate the point raised by the reviewer. Indeed, when comparing our EC_{REC-Apo} structure vs the Pol II pre-initiation complex bound to TFIIH (PDB 7NW0), we notice that RECQL5 helicase binding would sterically clash with one of the TFIIH helicase subunits, XPB. Therefore, as the reviewer points out, initiation factors such as TFIIH would likely not be able to form a complex with RECQL5 bound to Pol II. By contrast, overlapping our EC_{REC-Apo} structure with the Pol II EC bound with Elf1 and Spt4/5 (PDB 8JH2) shows that RECQL5 binding is compatible with the binding of the elongation factors Spt4 and Spt5, although showing a minor steric clash with the Elf1 elongation factor. We provide the structural comparisons in **Extended Data Fig. 6a,b** and discuss them briefly in the text (lines 213–221).

Reviewer #2:

The manuscript from the Nogales group is a tour de force, providing abundant, high-quality structural data which examine in detail the functional interactions between an elongating RNA polymerase II and the helicase RECQL5. RECQL5 has been shown previously to slow down transcription by RNAPII in vitro, and to be important in preventing DNA damage following replicative or transcriptional stress. The paper presents single-particle EM structures of an RNAPII elongation complex, alone and in combination with truncated RECQL5 in nucleotide-free, AMP-PNP and ADP forms. These structures show the binding determinants between RNAPII, RECQL5, and the nucleic acids. Combined with previous structural analyses of RNAPII conformations, these structures paint a vivid picture of the mechanism by which RECQL5 slows transcription and helps to bypass transcriptional blocks, while alternating between apo-, ATP and ADP-bound states.

The results are technically remarkable. The authors cleverly classify and refine individual molecular images from heterologous grids to achieve high resolution within the polymerase active site, allowing to precisely define the register of the polymerase on the RNA/DNA structure. The resolution of the helicase domains is sufficient to observe the position of bound ATP and the domain arrangement; Compared to the apo- and ATP structures, the observation of a less ordered state of the helicase domains when bound to ADP gains significance.

The conclusions from the structural analysis are convincing and the authors are careful not to over-interpret the results. The description and the figures are clear and well-explained-

I cannot really find any faults with this manuscript, and it should be published rapidly.

We thank the reviewer for their enthusiasm for our study and appreciate their complimentary remarks.

One small question I have: the DNA/RNA substrate used in the structural analysis includes a 3' single-stranded DNA extension on the non-transcribed strand, which is the docking site for binding of RECQL5; the binding mode seems to prefer ssDNA. Could the authors comment on what this ssDNA region may represent in a genomic environment, and whether there is any biochemical data to support a role of a ssDNA tail for the action of RECQL5.

We thank the reviewer for these important questions. As we discussed above, the single-stranded DNA region could represent several structures encountered in the genomic environment. First, it could represent the displaced single-stranded ntDNA in R-loops, which are formed as a result of transcription³. A given transcribing Pol II could encounter an R-loop created by the next downstream Pol II, whose single-stranded DNA region could be bound by RECQL5. Second, loading of RECQL5 at DSB sites through its interaction with the MRN complex⁴ and subsequent 3' to 5' helicase translocation until encountering an approaching Pol II could also lead to a local single-stranded region downstream of Pol II. Finally, we speculate that the single-stranded region in the replication fork leading strand at transcription-replication conflicts could also serve as a binding site for RECQL5 downstream of Pol II, from where it could translocate further upstream toward Pol II.

With respect to the second point, we have now confirmed through RNA extension assays that RECQL5 can inhibit Pol II transcription on our nucleic acid scaffold, which includes a downstream single-stranded extension. These results are shown in **Fig. 4h**, and support the notion that a single-stranded downstream ntDNA could serve as a platform for RECQL5 binding in order to inhibit Pol II.

References for Reviewer Response

1. Kassube et al. Structural mimicry in transcription regulation of human RNA polymerase II by the DNA helicase RECQL5. *Nat. Struct. Mol. Biol.* **20**, 892–899 (2013).
2. Kettenberger et al. Complete RNA Polymerase II Elongation Complex Structure and Its Interactions with NTP and TFIIS. *Mol. Cell* **16**, 955–965 (2004).
3. García-Muse & Aguilera, R Loops: From Physiological to Pathological Roles. *Cell* **179**, 604–618 (2019).
4. Zheng et al. MRE11 complex links RECQ5 helicase to sites of DNA damage. *Nucleic Acids Res.* **37**, 2645–2657 (2009).
5. Chappidi et al. Fork Cleavage-Religation Cycle and Active Transcription Mediate Replication Restart after Fork Stalling at Co-transcriptional R-Loops. *Mol. Cell* **77**, 528–541 (2020).

Reviewer #1:

The authors have addressed all of my concerns.

We thank the reviewer for their feedback during the revision process, which helped improve our paper.

Reviewer #2:

The revised manuscript has addressed the questions I raised and presents a much clearer view of the interactions of the polymerase and associated factors with the DNA/RNA. I recommend publication of the paper.

We thank the reviewer for their questions and positive evaluation of our paper.